

# From shear to veer: theory, statistics, and practical application

Mark Kelly[1] and Maarten Paul van der Laan[1]

[1]Department of Wind Energy, Danish Technical University, Risø Lab/Campus Frederiksborgvej 399, Roskilde 4000
**Correspondence:** Mark Kelly (MKEL@dtu.dk)

**Abstract.**

In the past several years, wind veer — sometimes called 'directional shear' — has begun to attract attention due to its effects on wind turbines and their production, particularly as the length of manufactured turbine blades has increased. Meanwhile, applicable meteorological theory has not progressed significantly beyond idealized cases for decades, though veer's effect on the wind speed profile has been recently revisited. On the other hand the shear exponent ($\alpha$) is commonly used in wind energy for vertical extrapolation of mean wind speeds, as well as being a key parameter for wind turbine loads calculations and design standards.

In this work we connect the oft-used shear exponent with veer, both theoretically and for practical use. We derive relations for wind veer from the equations of motion, finding the veer to be composed of separate contributions from shear and vertical gradients of cross-wind stress. Following from the theoretical derivations, which are neither limited to the surface-layer nor constrained by assumptions about mixing length or turbulent diffusivities, we establish simplified relations between the wind veer and shear exponent for practical use in wind energy. We also elucidate the source of commonly-observed stress-shear misalignment and its contribution to veer, noting that our new forms allow for such misalignment. The connection between shear and veer is further explored through analysis of one-dimensional (single-column) Reynolds-averaged Navier-Stokes solutions, where we confirm our theoretical derivations as well as the dependence of mean shear and veer on surface roughness and atmospheric boundary layer depth in terms of respective Rossby numbers.

Finally we investigate the observed behavior of shear and veer across different sites and flow regimes (including forested, offshore, and hilly terrain cases) over heights corresponding to multi-megawatt wind turbine rotors, also considering the effects of atmospheric stability. From this we find empirical forms for the probability distribution of veer during high-veer (stable) conditions, and for the variability of veer conditioned on wind speed. Analyzing observed joint probability distributions of $\alpha$ and veer, we compare the two simplified forms we derived earlier and adapt them to ultimately arrive at more universally applicable equations to predict the mean veer in terms of observed (i.e., conditioned on) shear exponent; lastly, the limitations, applicability, and behavior of these forms is discussed along with their use and further developments for both meteorology and wind energy.



## 1   Introduction

The shear exponent has generally not been used or accepted by meteorologists, as it does not (directly) relate to the physics of atmospheric flow, nor to the most important boundary condition—the surface. Regarding the latter, in contrast with similarity theory (Monin and Obukhov, 1954), the shear exponent does not contain explicit information about the surface roughness. However, the shear exponent can be related to surface properties in a generalized way, as well as to turbulent kinetic energy

and atmospheric stability (buoyancy) as shown by e.g. Kelly et al. (2014a). This is particularly useful above the atmospheric surface layer (ASL), where micrometeorological theory based on ASL assumptions fails—and where the effects of the surface are neither dominant nor simple enough to be characterized through accepted ASL parameterizations. As practiced in the wind energy resource assessment community for decades, the shear exponent can thus be preferable over similarity theory for use in vertical extrapolation (Irwin, 1979; Mikhail, 1985; Petersen et al., 1998) with quantification of uncertainty in its use more

recently reinforcing such (Triviño et al., 2017; Kelly et al., 2019b). Shear is also a key parameter for flow characterization towards loads simulations, being seen to systematically affect various turbine loads (e.g. Dimitrov et al., 2018; Robertson et al., 2019).

Veer has received much less attention than shear, though its potential importance to wind energy has been noted more recently. In the meteorological literature, where veer is often labelled as 'directional shear' or 'turning,' Markowski and Richard-

son (2006) reviewed the distinction between veer and vertical gradients of wind speed, listing studies of meteorological phenomena that considered veer (though they focused on convective storms). While some works in meteorology have investigated veer, these have tended to focus on the angular difference between winds at the top of the atmospheric boundary layer (ABL) and the surface (e.g. Clarke, 1975; Brown et al., 2005; Grisogono, 2011; Lindvall and Svensson, 2019),and are not generally suited for engineering applications. For wind energy, Murphy et al. (2020) looked at the veer (and shear) along with power pro-

duction measured over a six-month period, finding a minor but non-negligible effect of veer on power production for a utility scale turbine. Gao et al. (2021) found positive veer over the upper half of a single (2.5MW) clockwise-turning turbine rotor to reduce power production, opposite and slightly larger than the corresponding effects of negative veer there; they also showed the rotor's lower-half veer was less significant than the upper half. Shu et al. (2020) examined measurements from a lidar offshore between islands southwest of Hong Kong, observing larger veer when hilly terrain was upstream compared to more open

sea conditions; they also noted seasonal variations. For power production, the veer was incorporated into rotor-equivalent wind speed (REWS) by Choukulkar et al. (2016), whom found it to generally decrease production at two sites; Clack et al. (2016) found similar results from weather assimilation model output over the USA, along with higher production at night and lower power during daytime at most locations. Wind veer has also been examined with regard to its connection with the distortion and lateral movement turbine wakes via measurements and simulations (e.g. Abkar et al., 2018; Brugger et al., 2019), also

including yaw-misalignment affects (Hulsman et al., 2022; Narasimhan et al., 2022).

In this paper we elucidate analytical and statistical connections between a number of key parameters used to describe atmospheric boundary layer flow, with focus on the vertical variation of wind velocity. This follows the earlier work of Kelly



et al. (2014a) that gave forms for low-order statistics of shear exponent $\alpha$, relating $\alpha$ to turbulence intensity and stability; here we derive new relations for the turning of the wind in terms of shear, going beyond classical Ekman-type analysis.

## 2 Theory and development

### 2.1 Shear exponent

Just as potential temperature—the buoyancy variable commonly-used in meteorology—was labeled the "meteorologist's entropy" by Bohren and Albrecht (1998), one could call the shear exponent ($\alpha$) the "wind engineer's phi-function." Specifically this follows from the definition of shear exponent

$$\alpha \equiv \frac{\partial U/\partial z}{U/z} = \frac{\partial \ln U}{\partial \ln z} \tag{1}$$

and the dimensionless wind speed gradient

$$\Phi_m \equiv \frac{dU/dz}{u_{*0}/\kappa z} = \frac{\kappa U}{u_{*0}}\alpha; \tag{2}$$

used in meteorology, where $u_{*0}$ is the surface layer friction velocity (square root of kinematic shear stress), $\kappa = 0.4$ is the von Kármán constant, and $z$ is the height coordinate[1]. We remind that (1) is derived from the power-law expression for wind speed

$$\frac{U}{U_{\mathrm{ref}}} = \left(\frac{z}{z_{\mathrm{ref}}}\right)^\alpha, \tag{3}$$

which is assumed to be valid over some extent around height $z_{\mathrm{ref}}$, with $U_{\mathrm{ref}} \equiv U(z_{\mathrm{ref}})$. The power-law (3) with shear exponent (1) has been used in wind engineering for decades (e.g. Irwin, 1979; Petersen et al., 1998) due to its simplicity, and because it doesn't require any information other than the wind speed at two heights. Although (1) and (2) might appear to be quite alike, one can see a phenomenological difference when comparing the wind speed profiles resulting from these relations. In Monin-Obukhov ("M-O") theory $\Phi_m$ is a function of the stability $z/L$ which is proportional to surface heat flux $H_0$ divided by $u_{*0}^3$, i.e. the reciprocal Obukhov length is $1/L = \kappa(g/T_0)H_0/u_{*0}^3$ where $T_0$ is the background temperature and $g$ is the gravitational acceleration (Monin and Obukhov, 1954); the $\Phi_m$ function and corresponding M-O wind profile (which arises via integrating $dU/dz$ in (2) from a height equal to the roughness length $z_0$ up to height $z$) thus require a number of assumptions and more information than calculation of $\alpha$ via (1) or use of the power-law (3). Monin-Obukhov wind profiles also require the surface roughness length ($z_0$), while the friction velocity $u_{*0}$ (and thus shear stress) is assumed to be constant in the surface layer where M-O theory is most valid[2]; further, the assumptions of stationarity and a uniform flat surface are implicit in use of M-O theory. Following surface layer theory one could write an equivalent shear exponent $\alpha_{\mathrm{ASL}} = \Phi_m(z/L) / [\ln(z/z_0) - \Psi_m(z/L)]$ where

---

[1]The full derivative $(d/dz)$ is used in (2) because of the assumption of horizontal homogeneity by Monin-Obuhkov similarity theory from which $\Phi_m$ arises.

[2]The 'constant-flux layer' in surface-layer theory does not require exactly constant fluxes with height, as is often presumed. The label and assumption are that the *non-dimensional* fluxes, normalized by ABL scales, are constant with $z$ (Horst, 1999; Wyngaard, 2010); i.e., the ASL is the layer over which the decrease in $u_*^2$ is small compared to $u_{*0}^2$, roughly the bottom 10% of the ABL.



$\Psi_m = \int_{z_0}^{z}[1 - \Phi_m(z'/L)]d\ln z'$ is the M-O wind speed correction function; the analytic forms for $\Phi_m$ and $\Psi_m$ differ in stable and unstable conditions, and have been determined empirically in decades past (Businger et al., 1971; Carl et al., 1973; Li, 2021). But Monin-Obukhov similarity theory and its assumptions (such as constant $u_*$), as well as established forms for $\Phi_m$, fail above the surface layer;[3] this motivates use of $\alpha$ in applications such as wind energy, as (1) does not directly rely on surface-layer assumptions.

### 2.1.1 Relation to stability and turbulence

As shown by Kelly et al. (2014a), in horizontally homogeneous conditions the steady or mean balance of turbulent kinetic energy (TKE) can be written in terms of shear exponent as

$$\alpha = \frac{z}{U}\frac{(\varepsilon - B - T)}{-\langle uw \rangle} \tag{4}$$

for a given height $z$, where the streamwise direction is defined by the mean wind $U(z)$ and we have suppressed $z$-dependences for brevity; here $\langle uw \rangle$ is the turbulent horizontal momentum flux (kinematic stress), $T$ is the total (turbulent plus pressure) transport, $B$ is buoyant production, and $\varepsilon$ is the viscous dissipation rate of TKE. Within the ASL under these conditions where M-O theory is valid, using the neutral value of dissipation rate as $\varepsilon_0 \equiv u_{*0}^3/(\kappa z)$ along with the dimensionless functions $\Phi_\varepsilon \equiv \varepsilon/\varepsilon_0$ and $\Phi_T \equiv T/\varepsilon_0$ (Kaimal and Finnigan, 1994), we can express an ASL version of (4) as

$$\alpha_{\mathrm{ASL}} = \frac{u_{*0}}{\kappa U}\left(\Phi_\varepsilon + \frac{z}{L} - \Phi_T\right) \approx I_u\left(\Phi_\varepsilon + \frac{z}{L} - \Phi_T\right) \tag{5}$$

since by definition $B = -z/L$ and $u_{*0}^2 = -\langle uw \rangle$; here $I_u \equiv \sigma_u/U$ is the streamwise turbulence intensity. The dimensionless dissipation rate (M-O function) $\Phi_\varepsilon \geq 1$ is roughly $1 + 5z/L$ in stable conditions and increases more weakly with $-z/L$ in unstable conditions (Panofsky and Dutton, 1984; Kaimal and Finnigan, 1994); meanwhile the transport is negligible in stable conditions but $\Phi_T > 0$ in unstable conditions (e.g. Wyngaard, 2010). Thus in stable conditions ($L^{-1} > 0$) one can see $\alpha$ is larger than in neutral conditions, while in unstable conditions $\alpha$ becomes smaller. Above the ASL this will also generally be the case, though analytic nondimensional forms become difficult to derive, while the flow becomes affected by more terrain upwind and associated inhomogeneities; furthermore in stable conditions the local stability (at a given $z$) becomes increasingly more important than surface-based $z/L$ (Derbyshire, 1990). As will be shown below, the most common and mean conditions at contemporary rotor heights qualitatively follow (5), but due to these and other non-ideal effects (e.g. nonstationary transients) large deviations can occur. We note that in this work we are not searching for analytical forms for $\alpha$ or surface-layer behavior; rather, we are concerned with how $\alpha$ relates to the ***veer***, especially over heights corresponding to wind turbine rotors, a portion of which commonly extends beyond the ASL.

---

[3]We note that Kelly and Gryning (2010) adapted M-O theory to long-term means and Kelly and Troen (2016) extended this beyond the surface layer within the European Wind Atlas (WAsP) framework, thus addressing the stationarity and surface homogeneity aspects. However, the purpose and scope of the current article is to examine the commonly-used shear exponent and its connection with veer, not on vertical extrapolation methods per se.





## 2.2 Veer

For the simplified general case of Coriolis-affected mean flow, we write the horizontal mean velocity vector $\{U, V\}$ as a complex number, $S \equiv U + iV = |S|e^{i\varphi}$. For a mean wind direction defined at some height $z$, the veer can be defined as a directional shear $\partial\varphi/\partial z$ through the wind direction

$$\varphi(z) = \arg[S(z)] = \arctan\left[\frac{V(z)}{U(z)}\right]. \tag{6}$$

In most of the micrometeorological literature, the mean wind direction is defined based on the surface stress (i.e. via the winds closest to the surface, so $\varphi_0 \equiv \varphi(0) = 0$). We follow this convention unless stated otherwise, as done for some expressions later in section 2.3; one could also choose to define the coordinate system based on the geostrophic wind direction (e.g. Svensson and Holtslag, 2009).

As is classically known in micrometeorology (e.g. Hess and Garratt, 2002), the veer across the entire ABL depends primarily on the Coriolis parameter $f$ (thus latitude), geostrophic wind speed $|G|$, and surface roughness length $z_0$, but is also affected by the ABL depth $h$ and stability (as confirmed via Reynolds-averaged Navier-Stokes simulations by van der Laan et al., 2020). The veer across a fraction $\Delta z/h$ of the ABL will also depend on these parameters; thus for a given site and height, $\Delta\varphi/\Delta z$ will have a distribution due to variations in these parameters.This will become clearer below as we examine the relationship between veer and shear.

The Coriolis-affected mean momentum balance can be written in the form

$$\frac{\partial S}{\partial t} = 0 = -if(S - G) - \frac{\partial\langle sw\rangle}{\partial z} \tag{7}$$

for stationary and horizontally homogeneous conditions (thus neglecting advection). Here the kinematic horizontal pressure gradient $\nabla p/\rho = f\{V_G, -U_G\}$ is also written like a velocity in complex form as $G \equiv U_G + iV_G = (-\partial p/\partial y + i\partial p/\partial x)/(\rho f)$. The mean stresses are dominated by vertical momentum transport $\langle sw\rangle$, where $w$ denotes (turbulent) vertical velocity fluctuations and $s \equiv u + iv$ the horizontal velocity fluctuations.

At a given height $z$, taking the differential of (6) (recalling $d\arctan x = dx/[1 + x^2]$ and using the chain rule) gives

$$d\varphi = \frac{U\,dV - V\,dU}{|S|^2} = i\frac{S^*dS - SdS^*}{2|S|^2} = -i\,d\ln\left(\frac{S}{|S|}\right); \tag{8}$$

here the superscript asterisk denotes complex conjugate. Applying $\partial/\partial z$ to (8) and (7) and combining provides a basic expression for veer:

$$\frac{\partial\varphi}{\partial z} = \frac{U}{|S|^2}\left[\frac{1}{f}\frac{\partial^2\langle uw\rangle}{\partial z^2} + \frac{\partial V_G}{\partial z}\right] + \frac{V}{|S|^2}\left[\frac{1}{f}\frac{\partial^2\langle vw\rangle}{\partial z^2} - \frac{\partial U_G}{\partial z}\right]. \tag{9}$$

In the case of zero geostrophic shear ($d\mathbf{G}/dz = 0$), if the coordinate system's $x$-axis is defined by the mean wind direction at the height $z$ where the veer is sought, then (9) can be written more simply as

$$\left.\frac{\partial\varphi(z)}{\partial z}\right|_{d\mathbf{G}/dz \to 0} = \frac{1}{f|S|}\frac{\partial^2}{\partial z^2}\langle uw\rangle|_{\mathbf{e}_x\|\mathbf{U}(z)}. \tag{10}$$





Though (9) and (10) are not directly very useful for relating veer to shear, they illustrate that the *curvature* of stress profiles
and Coriolis effect are the basis for mean veer following (7), and also that geostrophic shear can further contribute to veer (e.g.
due to baroclinity, Hoxit, 1974; Arya and Wyngaard, 1975; Pedersen et al., 2013).

## 2.3 Relating veer to shear

Towards relating the veer to shear one can alternately take the time derivative of (8); using the real and imaginary parts of (7),
in the horizontally homogeneous limit (ignoring advection) one obtains a rate equation for mean wind direction:

$$\frac{\partial \varphi}{\partial t} = \left[ \frac{V}{|S|^2} \frac{\partial \langle uw \rangle}{\partial z} - \frac{U}{|S|^2} \frac{\partial \langle vw \rangle}{\partial z} \right] + f \left( \frac{|G|}{|S|} \cos \gamma - 1 \right). \tag{11}$$

The 'turning' angle $\gamma \equiv \varphi - \varphi_G$ between geostrophic and mean wind directions (e.g. Wyngaard, 2010) arises through[4]

$$U_G U + V_G V = \mathbf{U} \cdot \mathbf{G} = |S||G| \cos \gamma$$

by taking $\partial/\partial t$ of (6) or equivalently $U \partial V/\partial t - V \partial U/\partial t$ via (8). The geostrophic wind direction is defined as $\varphi_G \equiv \arctan(V_G/U_G)$,
and the 'cross-isobar' angle, i.e. the turning over the whole ABL ($\gamma_0 = \varphi_0 - \varphi_G$), is generally less than $45°$ (Grisogono, 2011)[5];
in a right-handed coordinate system, regardless of whether $\mathbf{x}$ is chosen to align with $\mathbf{G}$ or the surface-layer wind velocity $\mathbf{U}_{\text{ASL}}$,
the turning tends to $\gamma > 0$ in the Northern hemisphere[6]. We remind that $\varphi$, and thus $\gamma$, can vary with height $z$ (as can $\varphi_G$ in
baroclinic conditions).

Assuming statistical stationarity so that $\partial \varphi/\partial t = 0$, the vertical derivative of (11) can be written most conveniently in terms
of the dimensionless deviation of the wind from streamwise; taking the vertical derivative of (11) if we again take $d\mathbf{G}/dz = 0$
(neglect baroclinity), then

$$\frac{\partial \cos \gamma}{\partial z} = \frac{1}{|G|} \frac{\partial |S|}{\partial z} + \frac{1}{f|G|} \frac{\partial}{\partial z} \left[ \frac{U}{|S|} \frac{\partial \langle vw \rangle}{\partial z} - \frac{V}{|S|} \frac{\partial \langle uw \rangle}{\partial z} \right]. \tag{12}$$

As it is expressed in terms of angular *differences* $\gamma$, the equation above is independent of whether the coordinate system is
defined at the surface or by the geostrophic wind. Expression (12) clearly separates the shear and Coriolis/stress contributions
to veer. However, it can be simplified, and is most meaningful, if the coordinate system is defined at the height $z$ for which
it is applied; in practice the veer is typically calculated around hub height, or from hub to tip, or between measurement and
hub heights. Re-expressing (11) with the coordinate system defined by having $x$ in the mean wind direction at height $z$, so that
$\mathbf{S}(z) = U(z)\mathbf{e}_x$ and $|S(z)| = U(z)$, in the mean (for $d\varphi/dt = 0$) one has

$$\cos \gamma = \frac{|S|}{|G|} + \frac{1}{f|G|} \frac{\partial \langle vw \rangle_\perp}{\partial z} \tag{13}$$

---

[4]The turning angle can also be expressed in complex notation, recalling that the angle between vectors written in complex notation (here $\mathbf{U} \to S$ and
$\mathbf{G} \to G$) can be recovered by taking $\Re\{G^* S\}$, i.e. $|G||S|\Re\{e^{-i(\varphi+\gamma)} e^{i\varphi}\} = |G||S| \cos \gamma$.

[5]The ABL turning angle $\gamma_0$ cannot exceed $45°$, according to the Ekman equations (or their numerical solution, as in van der Laan et al., 2020). However, in
some situations, which tend to involve horizontal inhomogeneities, $\gamma_0 > 45°$; these include e.g., baroclinity, terrain-induced turning (especially with stability),
convective cells, and various persistent storm structures.

[6]In the Southern hemisphere, the signs are reversed: geostrophic flow around a local low-pressure moves clockwise, with surface-induced turbulence
('friction') causing the flow to again increasingly turn towards low pressure as the surface is approached, and thus $\gamma < 0$.





where we use the shorthand notation $\langle vw \rangle_\perp$ to denote the stress perpendicular to the mean flow at a given height. Taking the inverse cosine and subsequently the vertical derivative, noting that $\partial\gamma/\partial z = \partial\varphi/\partial z$ and $d\arccos x = -dx/\sqrt{1-x^2}$ while recalling $\partial|S|/\partial z = \alpha|S|/z$, we get

$$\left.\frac{\partial\varphi}{\partial z}\right|_{\mathbf{e}_x\|\mathbf{U}(z)} = -\left(\frac{|S|}{|G|}\frac{\alpha}{z} + \frac{1}{f|G|}\frac{\partial^2\langle vw\rangle_\perp}{\partial z^2}\right)\left[1 - \left(\frac{|S|}{|G|} + \frac{1}{f|G|}\frac{\partial\langle vw\rangle_\perp}{\partial z}\right)^2\right]^{-1/2}. \tag{14}$$

The more generic form of this, for an arbitrary coordinate system, follows from (11):

$$\frac{\partial\varphi}{\partial z} = \frac{-\frac{|S|}{|G|}\frac{\alpha}{z} + \frac{1}{f|G|}\frac{\partial}{\partial z}\left(\frac{V}{|S|}\frac{\partial\langle uw\rangle}{\partial z} - \frac{U}{|S|}\frac{\partial\langle vw\rangle}{\partial z}\right)}{\sqrt{1 - \left[\frac{|S|}{|G|} - \frac{1}{f|G|}\left(\frac{V}{|S|}\frac{\partial\langle uw\rangle}{\partial z} - \frac{U}{|S|}\frac{\partial\langle vw\rangle}{\partial z}\right)\right]^2}}. \tag{15}$$

We note that (14) and (15) are more direct alternatives to dealing with functions of $\varphi_G$, which become apparent if one expands $\cos\gamma$ in (12) or (13). However, one can see that there can be an angular dependence within the stress-related parts written above; when considered in coordinates defined with the $x$-direction aligned with the mean wind at height $z$, in the general forms (12) and (15), $U/|S|$ and $V/|S|$ can be written as $\cos\varphi$ and $\sin\varphi$, respectively. From (12) and using $\cos\gamma = \cos\varphi\cos\varphi_G + \sin\varphi\sin\varphi_G$, then in coordinates again defined by $|S(z)| = U(z)$, after some rearranging we arrive at an expression for veer like (14):

$$\frac{\partial\varphi}{\partial z} = \frac{\frac{|S|\alpha}{|G|z} + \frac{\partial^2\langle vw\rangle_\perp/\partial z^2}{f|G|}}{\sin\varphi_G + \frac{\partial\langle uw\rangle_\|/\partial z}{f|G|}}. \tag{16}$$

Compared to (14) this lacks a negative sign, but $\sin\varphi_G$ is negative and with larger magnitude than the positive contribution to the denominator, $\partial\langle uw\rangle_\|/\partial z/(f|G|)$; this will become more apparent in the sections which follow. We also remind that in these coordinates $\varphi_G = \gamma_G(z)$, and opposite signs will occur for the southern hemisphere (expressions 14–16 give $d\varphi/dz$ signed for the northern hemisphere in mathematical coordinates, reflecting winds rotating on average clockwise with increasing height, i.e. negative veer).

For wind energy $\partial(\cos\varphi)/\partial z$ might be considered as relevant as $\partial\varphi/\partial z$, because it allows direct expression of the veer-induced variation in streamwise wind velocity component relative to a reference height such as hub height. One could expect that the reduction of $\cos\varphi$ away from a given $z$ counteracts the effect of typically positive shear; if desired, the veer can be simply re-expressed later in terms of $\cos\varphi$ for a given coordinate system, instead of trying to use an expression such as (12).

### 2.3.1 Misalignment of shear and stress

One can see a connection between the shear, veer, and stress in (9) and (12), and we can further examine the relation between shear and stress using complex notation as in (7). The 'misalignment' can be expressed via the angle between $\partial S/\partial z$ and $\langle sw \rangle$, i.e.

$$\beta_{\mathrm{ma}} \equiv (\varphi - \varphi_{sw}) = \arg(S) - \arg(\langle sw \rangle). \tag{17}$$





The root of such misalignment arises in the rate-equation for $\langle sw \rangle$. In the limit of horizontal homogeneity, if we combine the stress budgets (e.g. see Wyngaard, 2010), i.e. adding $\partial\langle uw\rangle/\partial t$ to $i\partial\langle vw\rangle/\partial t$, we may write

$$\frac{\partial\langle sw\rangle}{\partial t} = 0 \simeq \langle w^2\rangle\frac{\partial S}{\partial z} - \frac{\langle sw\rangle}{\tau_R} - \frac{\partial}{\partial z}\langle sww\rangle. \tag{18}$$

The pressure-strain contribution has been written as $\langle sw\rangle/\tau_R$ via the commonly-used Rotta parameterization, where $\tau_R$ is the Rotta time scale; this is the basis for commonly used flux-gradient relations (Wyngaard, 2004). In such mixing-length relations, i.e. using the 'Boussinesq hypothesis,' $\langle w^2\rangle\tau_R$ is simply written as a turbulent diffusivity $-\nu_T$, and the final term in (18) is neglected. We continue to neglect advection and horizontal transport (such as $U\partial\langle sw\rangle/\partial x$ and $\partial\langle suw\rangle/\partial x$ respectively); these can also contribute to misalignment between $\partial S/\partial z$ and $\langle sw\rangle$ in areas of upwind horizontal inhomogeneity such as nonuniform terrain and turbine wakes. Thus in models where an eddy-diffusivity (flux-gradient relation) is used, such as most RANS solvers which employ 2-equation turbulence models, for flow over homogeneous surfaces there will be no stress-shear misalignment.

Ghannam and Bou-Zeid (2021) derived a dimensionless relation in terms of the angular differences $\beta_{\mathrm{ma}}$ and $\gamma$ instead of velocity components; although it does not afford convenient description of the veer, it can be re-cast to show the effect of the misalignment angle:

$$f|G|\sin\gamma = -\frac{\partial|\langle sw\rangle|}{\partial z}\cos\beta_{\mathrm{ma}} - |\langle sw\rangle|\sin\beta_{\mathrm{ma}}\frac{\partial\varphi_{sw}}{\partial z}. \tag{19}$$

Thus when the stress is aligned with the shear ($\beta_{\mathrm{ma}} = 0$), then $f|G|\sin\gamma = -\partial|\langle sw\rangle|/\partial z$; this can be seen as a case of (13). The contribution of stress-shear misalignment to the veer can also be seen considering (18) with our earlier derivations, with misalignment modifying the stresses. For example the cross-wind stress in (13)–(15) can be written

$$\langle vw\rangle_\perp = -\nu_T\left[\frac{\partial V_\perp}{\partial z} + \frac{\partial\langle vww\rangle_\perp/\partial z}{2k/3}\right] \tag{20}$$

since the Rotta timescale can be expressed in terms of turbulent kinetic energy $k$ via $\nu_T = \tau_R\langle uu+vv+ww\rangle/3$ (see Pope, 2000; Hatlee and Wyngaard, 2007). But the turbulent third-order moment $\langle sww\rangle$ is difficult to measure, so a model for it would be needed in order to *explicitly* incorporate misalignment into veer predictions. Fortunately the misalignment $\beta_{\mathrm{ma}}$ tends to be small in the surface layer (Geernaert, 1988), and also beyond the surface layer over homogeneous terrain or long fetch over water, especially without baroclinity (Berg et al., 2013). However, it has been known for decades (Moeng and Wyngaard, 1989) that turbulent transport is relevant in convective ABLs, so one expects more misalignment in unstable conditions; indeed Santos et al. (2021) saw this from measurements over multiple heights over a land and sea site, as did Berg et al. (2013) to a lesser extent (due to the relatively short measurement campaign) over water. The misalignment tends to be smaller in neutral conditions, and thus we do not (yet) offer explicit treatment for it.

### 2.3.2 Alignment of shear and stress; canonical solutions

When turbulent transport of stress is negligible (along with baroclinity and inhomogeneity), in steady conditions the stress and mean velocity gradient are aligned, allowing use of an eddy diffusivity. The veer can then be cast as a nonlinear differential





equation, which in flow-following coordinates at height $z$ is

$$\frac{\partial \varphi}{\partial z} = \frac{-1}{f|G|} \frac{\partial^2}{\partial z^2} \left( \nu_T(z) \frac{\partial U}{\partial z} \right). \tag{21}$$

This defies analytical solution, though one can note limits of the veer by considering two canonical cases where it can be solved: the Ekman and Ellison regimes, corresponding to simple prescriptions for $\nu_T$. Such limits were considered by van der Laan et al. (2020) for the geostrophic drag coefficient $c_G \equiv u_*/|G|$ and ABL-integrated veer (cross-isobar angle) $\gamma_0 \equiv \varphi_0 - \varphi_G$.

*Ekman solution.*

Ekman (1905) assumed the turbulent stress was related to the mean shear using a constant eddy-viscosity $\nu_{\mathrm{Ek}}$, which in our notation is expressible as $\langle sw \rangle = -\nu_{\mathrm{Ek}} \partial S / \partial z$. Thus the momentum balance (7) simplifies to

$$f(S - G) = -i \nu_{\mathrm{Ek}} \frac{\partial^2 S}{\partial z^2}, \tag{22}$$

which gives the classic Ekman solution

$$S_{\mathrm{Ek}} = G \left( 1 - e^{-(1+i)z/h_{\mathrm{Ek}}} \right), \tag{23}$$

where the characteristic Ekman ($e$-folding) height $h_{\mathrm{Ek}}$ is defined as $h_{\mathrm{Ek}} \equiv \sqrt{2\nu_{\mathrm{Ek}}/f}$. Simpler than relating Ekman veer to shear, the solutions above along with (9) give the veer directly as

$$\begin{aligned}
\frac{\partial \varphi_{\mathrm{Ek}}}{\partial z} &= \frac{-\nu_{\mathrm{Ek}}}{f|S_{\mathrm{Ek}}|^2} \left[ U \frac{\partial^3 U}{\partial z^3} + V \frac{\partial^3 V}{\partial z^3} \right] = \frac{e^{-z/h_{\mathrm{Ek}}}}{h_{\mathrm{Ek}}} \cdot \frac{\cos(z/h_{\mathrm{Ek}}) - \sin(z/h_{\mathrm{Ek}}) - e^{-z/h_{\mathrm{Ek}}}}{1 - 2e^{-z/h_{\mathrm{Ek}}} \cos(z/h_{\mathrm{Ek}}) + e^{-2z/h_{\mathrm{Ek}}}} \\
&\simeq \frac{-0.5 + z/(6h_{\mathrm{Ek}})}{h_{\mathrm{Ek}}};
\end{aligned} \tag{24}$$

this result has units of radians/m measured counter-clockwise, with the linear approximation[7] deviating from the exact form by less than 1% for $z < 1.5 h_{\mathrm{Ek}}$. Integrated over $z \pm \Delta z/2$, this gives the veer across an extent $\Delta z$:

$$\Delta \varphi_{Ek} \simeq \frac{-\Delta z}{2 h_{\mathrm{Ek}}} \left( 1 - \frac{\Delta z}{3 h_{\mathrm{Ek}}} \right). \tag{25}$$

The Ekman forms might be seen as an upper limit on veer for $h_{\mathrm{Ek}}$ on the order of typical ABL depths ($\sim$300–1000 m), analogous to what was found by van der Laan et al. (2020) for the cross-isobar angle $\gamma_0$.

From (23) one can also find an expression for the Ekman shear exponent $\alpha_{\mathrm{Ek}}$ via (1),

$$\alpha_{\mathrm{Ek}} = \frac{|\partial S_{\mathrm{Ek}}/\partial z|}{|S_{\mathrm{Ek}}|/z} = \frac{\sqrt{2}(z/h_{\mathrm{Ek}})}{\sqrt{1 - 2\cos(z/h_{\mathrm{Ek}})e^{z/h_{\mathrm{Ek}}} + e^{2z/h_{\mathrm{Ek}}}}} \simeq \left( 1 - \frac{\sqrt{2}}{\pi} \frac{z}{h_{\mathrm{Ek}}} \right). \tag{26}$$

This may also be seen as an upper limit, particularly in the surface layer where an unrealistically large diffusivity is assumed; one can see that Ekman theory predicts $\alpha \to 1$ approaching the surface.

*Linear diffusivity profile: 'modified Ekman' or surface-layer regime*

---

[7]The approximation is found by series expansion in $z/h_{\mathrm{Ek}}$ about 0; the same result is obtainable by taking the vertical derivative of (6), i.e. $\partial \left[ \arctan \left( \Im\{S_{\mathrm{Ek}}\}/\Re\{S_{\mathrm{Ek}}\} \right) \right] / \partial z$.





Using a surface-layer eddy-viscosity relation $\nu_T(z)=\kappa u_* z$ consistent with ASL theory, Ellison (1956) derived the solution

of (7), resulting in a profile of wind vector 'deficit' expressible as (Krishna, 1980)

$$S(z) - G = \frac{-2u_*}{\kappa}\left[\ker_0\left(\sqrt{\frac{2fz}{\kappa u_*}}\right) + i\,\mathrm{kei}_0\left(\sqrt{\frac{2fz}{\kappa u_*}}\right)\right]. \tag{27}$$

where $\ker_0(x)$ and $\mathrm{kei}_0(x)$ are the so-called Kelvin functions (see e.g. Abramowitz and Stegun, 1972). But the Ellison solution can be written more compactly and conveniently, similar to (23) with a complex argument, as

$$S(z) = G - \frac{2u_*}{\kappa}K_0\left(\sqrt{\frac{2ifz}{\kappa u_*}}\right) = G\left[1 - 2\frac{c_G}{\kappa}K_0\left(\frac{(1+i)z}{\sqrt{\nu_T(z)/f}}\right)\right] = G\left[1 - 2\frac{c_G}{\kappa}K_0\left(\sqrt{\frac{2iz}{h_{\mathrm{mE}}}}\right)\right]; \tag{28}$$

$K_0(x)$ is the zeroth-order modified Bessel function of the second kind, and the modified-Ekman length scale is defined by $h_{\mathrm{ME}} \equiv \kappa u_*/f$, also equal to $\nu_T(z)/fz$. For the range $0.02 \lesssim c_G \lesssim 0.06$ encountered in nature under neutral conditions (Hess and Garratt, 2002; van der Laan et al., 2020), for $z_H/h_{\mathrm{mE}} \gg 0.1$ the arctangent of $\Im\{S\}/\Re\{S\}$ can be approximated via series expansions of (27) or (28) to yield the practical result

$$\Delta\varphi(z_H, \Delta z) \approx \pi c_G \exp\left(-\sqrt{z'/z_H}\right)\Big|_{z_H - \Delta z/2}^{z_H + \Delta z/2}; \tag{29}$$

this follows the numerical solution to within $\sim 20\%$ for $0.3 \lesssim z/h_{\mathrm{mE}} \lesssim 2$, moreso for $c_G$ approaching 0.04.

It was shown in van der Laan et al. (2020) that the Ekman and Ellison solutions basically gave upper and lower limits, respectively, to observed full-ABL turning ($\varphi_G - \varphi_0$). Following this, in Fig. 1 we present veer profiles along with the relationship between veer and shear, for the Ekman and Ellison solutions; the former is calculated via the expressions in (24) and (26), while the latter is obtained via (28).

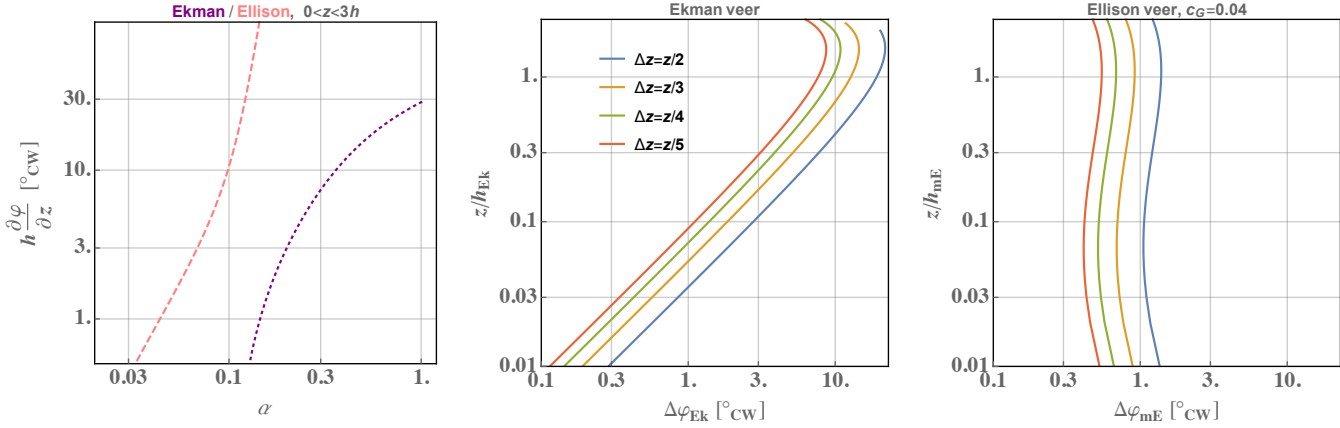

**Figure 1.** Veer behavior (plotted as degrees clockwise) for analytical/limiting cases of Ekman and Ellison. Left: veer versus shear exponent, for any Ekman or Ellison ABL depth $h$; Ekman is dotted purple, Ellison is dashed pink. Center/right: profiles of bulk veer for different $\Delta z$; Ellison solution in right-hand plot is numerical solution (without approximation).



One can see in Fig. 1 that the Ekman solution produces effectively less mixing away from the surface (for $z > \nu_{\mathrm{Ek}}/(\kappa u_*)$), and consequently a higher shear exponent than Ellison's. Similarly, the dimensionless Ekman veer exceeds that predicted by the Ellison solution, for $z/h_{\mathrm{Ek}} \gtrsim 0.1$, consistent with $\gamma_{0,\mathrm{Ek}} = 45°$, which exceeds the $\gamma_0$ of 5–15° predicted by Ellison's (van der Laan et al., 2020). However, we note that the depth $h$ can differ for Ekman and Ellison solutions; $h_{\mathrm{Ek}} = h_{\mathrm{mE}}$ only if one chooses $\nu_{\mathrm{Ek}} = (\kappa u_*)^2/2f$. We also point out that for larger $c_G$, i.e. smaller $\mathrm{Ro}_0$ (not shown in figure), near the surface

($z/h_{\mathrm{mE}} \lesssim 0.1$) Ellison's veer grows yet larger than the peak value shown at $z \approx 1.5h_{\mathrm{mE}}$ and relative to the behavior seen for $c_G = 0.04$; however, this idealized near-surface behavior is likely not relevant for wind applications.

## 2.4   Practical forms and application

To use the expressions derived for veer earlier, one needs the vertical derivatives of stress (or its profile) and the geostrophic wind speed; in particular the first and second vertical derivatives of the cross-wind stress $\langle vw \rangle_\perp$ appear in (14) and (16), along

with $|G|$. In wind energy applications, engineers typically lack site-specific stress profiles, unless they are taken from flow modelling; if the latter is reliable, then there is probably less need for the shear-based estimates for veer given in this work. The large-scale horizontal pressure gradients which drive ABL flow, expressible as the geostrophic wind $G$, are likewise rarely measured (though lidar measurements above the ABL can make this possible, e.g. Pedersen et al., 2013). The shear contribution to veer is multiplied by $|S/G|$ in (14)–(16). To obtain a practical form relating shear to veer, we can start by parameterizing

$|S|/|G|$; fortunately $|G|$ is commonly calculated in practice using a geostrophic drag law ('GDL', Rossby and Montgomery, 1935). Long used in wind applications such as WAsP (Troen and Petersen, 1989) and related wind resource software, it is expressible in scalar form as

$$|G| = \frac{u_*}{\kappa} \sqrt{\left[\ln\left(\frac{u_\star}{fz_0}\right) - A\right]^2 + B^2} \tag{30}$$

with components

$$\sin(\varphi_G - \varphi_0) = -B\frac{u_*}{\kappa|G|} \qquad \text{and} \qquad \cos(\varphi_G - \varphi_0) = \frac{u_*}{\kappa|G|}\left[\ln\left(\frac{u_*/f}{z_0}\right) - A\right] \tag{31}$$

where the empirical coefficients $\{A, B\}$ are assumed to be constants in typical wind application. The geostrophic drag coefficient $c_G \equiv u_*/|G|$ and ABL turning (cross-isobar angle) $\varphi_G$ are seen to vary with surface-Rossby number $\mathrm{Ro}_0$ (Blackadar and Tennekes, 1968); these and $\{A, B\}$ have been shown to depend on dimensionless stability $L^{-1}u_*/f$ (Arya, 1978; Kelly and Troen, 2016), strength of ABL-capping inversion (Zilitinkevich and Esau, 2002), and baroclinity (Arya and Wyngaard,

1975; Nieuwstadt, 1984). For practicality, we start by assuming near-neutral stability, which is appropriate in the mean for most places, as it represents by far the most frequently observed conditions (Kelly and Gryning, 2010); we continue to neglect baroclinity; and we neglect influence of the capping inversion strength.[8] With such assumptions, one can also write an

---

[8]We note Zilitinkevich and Esau (2005) gave a form for the GDL incorporating all three of these effects, and Liu et al. (2021) practically simplified that form, using LES to find its empirical constants in the case of nonzero effect of capping inversion strength per Coriolis parameter. However, the extra parameters needed are additional to what is required for the current theory given for climatological-mean conditions, and well beyond what is measured in practice.





(approximate) 'reverse' form of (30) to get the drag coefficient as (Troen and Petersen, 1989)

$$c_G \simeq \frac{c_{\mathrm{rGDL}}}{\ln \mathrm{Ro}_0 - A} \tag{32}$$

where the surface Rossby number is $\mathrm{Ro}_0 \equiv |G|/(fz_0)$ and $c_{\mathrm{rGDL}}$ is taken to be 0.485 following its use in the wind resource program WAsP for several decades. Alternate forms of (32) exist, such as that of Hess and Garratt (2002); the latter corresponds simply to setting $A = 1.28$ and $c_{\mathrm{rGDL}} = 0.472$ in (32). For a given roughness length $z_0$ and measured wind speed $|S|$, lacking the (surface) friction velocity $u_*$, one needs a relation to connect $u_*$ and $|S|$, in order to get $|G|$. This can be done through the same wind profile relation upon which the GDL is built, i.e. the log-law; one can use $u_* = \kappa |S|/\ln(z/z_0)$ within (30) or

alternately $|S|/|G| = (c_G/\kappa)\ln(z/z_0)$ using (32), where in the latter (30) is also employed to find $|G|$ within $\mathrm{Ro}_0$.

In practice one would like a direct estimate for the veer, using the routinely-measured shear, since $\alpha$ is seen to drive $\partial\varphi/\partial z$. One way could be to just ignore the stress divergence terms in (14) or (16), which with calculation of $|G|$ mentioned just above considerably simplifies the problem. However, this might not be justified, particularly if $u_*^2/(fh)$ is not negligible compared to $|S|$, as seen from comparing contributions to (14)–(16); this can be seen using the scaling $\partial\langle uw\rangle/\partial z \approx u_*^2/h$ where $h$ is the

ABL depth (e.g. Wyngaard, 2010). Thus we consider estimating vertical derivatives of the stresses, starting with the $\partial\langle uw\rangle/\partial z$ just mentioned, which can be used in (16). Similarly, one can estimate $\partial^2\langle vw\rangle/\partial z^2 \approx c_{vw}u_*^2/h^2$ or

$$\frac{\partial^2\langle vw\rangle/\partial z^2}{f|G|} \approx c_{vw}\frac{u_*^2}{f|G|h^2} = c_{vw}\frac{c_G^2}{h}\mathrm{Ro}_h \tag{33}$$

where $\mathrm{Ro}_h \equiv G/(fh)$ is the Rossby number based on ABL depth and $c_{vw}$ is of order 1; we will treat $c_{vw}$ as an empirical constant which is tuned later below. To use (16) we also need to find $\sin\varphi_G$; employing (31) and using trigonometric identities

to expand $\sin(\varphi - \varphi_0)$, with some rearrangement one obtains

$$\sin\varphi_G = \frac{-c_G}{\kappa}\left\{B - \left[\ln\left(\frac{u_*}{fz_0}\right) - A\right]\sin\varphi_0\right\}. \tag{34}$$

Employing this, (33), and $\partial\langle uw\rangle/\partial z \approx u_*^2/h$, along with with (30) or (32), allows one to then use (16).

On the other hand, using (14) is simpler and more convenient than (16), because it only requires $\partial\langle vw\rangle/\partial z$ in addition to the second derivative of $\langle vw\rangle$ just approximated in (33) above, so one can also simply approximate $\partial\langle vw\rangle/\partial z \approx h\partial^2\langle vw\rangle/\partial z^2$

and use (33); the GDL forms (30) and (32) then allow one to get $\mathrm{Ro}_h$ and $c_G$, respectively. Whether using (16) or (14), we note that the shear contribution to veer includes a surface-Rossby number ($\mathrm{Ro}_0$) dependence through $S/|G|$, while the stress-Coriolis contribution includes an ABL-depth dependence, $\mathrm{Ro}_h$; either way, if we do not neglect the latter, then we also need an estimate for the ABL depth $h$. If the shear contribution is expected to dominate variations in veer, then the estimate of $h$ may not be so crucial; we will consider this further below in our comparison with real-world cases, and also direct interested

readers to e.g. Liu and Liang (2010) for statistics of $h$ in different conditions.



## 3 Analysis and Discussion

This section presents analysis of results from RANS simulations of the neutral atmospheric boundary layer[9], and of observations at different sites (which include the impacts of stability). The simulations are analyzed to check the relations given here, as well as examine the behavior of and contributions to veer across the range of Rossby numbers ($Ro_0$ and $Ro_h$) encountered

in nature. Investigation of observations, spanning turbine rotor heights in five places having different wind regimes and conditions, includes probing the interconnected behaviors of shear (exponent) and veer with atmospheric stability – as well as their joint statistics, universal trends, and variation with wind speed. The statistical demonstration of observations is accompanied by predictions of veer using empirically updated forms of the relations given in the previous section, as well as the forms themselves.

### 3.1 RANS simulations of neutral ABLs

#### 3.1.1 Model and setup

The Navier-Stokes solver Ellipsys1D (van der Laan and Sørensen, 2017), which is a one-dimensional version of the multiblock general CFD solver Ellipsys3D (Sørensen, 1995), was used to simulate the Reynolds-averaged flow in neutral atmospheric boundary layers, including Coriolis forces. Assuming zero vertical velocity and constant pressure gradients, it solves the RANS

equations for incompressible flow with a finite-volume scheme. The ABL 'top' (above which turbulence is extinguished) is modelled via the length-scale limiter model of Apsley and Castro (1997) implemented into the $k$-$\varepsilon$ turbulence closure equations solved by Ellipsys1D, as outlined in van der Laan et al. (2020); this includes use of small ambient values of turbulence intensity and dissipation rate above the ABL, with $k$-$\varepsilon$ constants $C_\mu = 0.03$, $C_{\varepsilon 1} = 1.21$, $C_{\varepsilon 2} = 1.92$, $\sigma_k = 1.0$ and $\sigma_\varepsilon = 1.3$. The $k$-$\varepsilon$ model provides the stresses occuring in the RANS equations, via the flux-gradient relation and $\nu_T = C_\mu k^2/\varepsilon$; thus we see that

such turbulence closure gives stresses aligned with velocity gradients.

The domain height is set to $10^5\,\mathrm{m}$ to ensure it is much larger than $h$ for all simulations, and the bottom boundary is handled by a rough-wall condition (Sørensen et al., 2007). The numerical 'grid' is a vertical line, with the bottom cell height being $1\,\mathrm{cm}$ (placed above the roughness length) and the cells' sizes growing progressively upward with an expansion ratio of 1.2; the total number of cells is 384. At the bottom cell a Neumann condition is set for $k$ ($dk/dz = 0$) and $\varepsilon$ is set to the logarithmic value,

the wall stress is consequently defined by the neutral surface layer for this cell. More details, including a grid-refinement study, may be found in van der Laan et al. (2020).

Using a constant geostrophic wind speed, the flow is driven by a constant pressure gradient, starting with an initial wind profile set to $|G|$ at all heights; the ABL depth grows upward until convergence occurs, providing a steady solution and $h$ for a given choice of $z_0$, pressure gradient (thus $G$ and $f$), and turbulence ($k$-$\varepsilon$) limiting lengthscale $\ell_{\max}$. The Buckingham Pi

theorem can be used to reduce the four parameters $\{z_0, G, f, \ell_{\max}\}$ into two dimensionless groups, namely Rossby numbers for

---

[9]The neutral RANS simulations can also be translated into equivalent stable cases within the $k$-$\varepsilon$-$\ell_{\max}$ turbulence closure framework of Apsley and Castro (1997), following van der Laan et al. (2020).



$z_0$ and $\ell_{\max}$; for lengthscale-limited $k$-$\varepsilon$ RANS in the neutral ABL, one further has the relation

$$h = \ell_{\max}^{0.6} \left( \frac{|G|}{|f|} \right)^{0.4} \tag{35}$$

thus giving us the two Rossby numbers $\mathrm{Ro}_0$ and $\mathrm{Ro}_h$ for describing flow cases (van der Laan et al., 2020). Simulations were done over the full range of ABL depths, surface roughnesses, and wind speeds encountered in nature, which correspond to a

range of Rossby numbers spanning $10^5 < \mathrm{Ro}_0 < 10^{10}$ and $15.8 < \mathrm{Ro}_h < 661$. For simplicity $|G|$ was set to $10\,\mathrm{m\,s^{-1}}$ and $f$ to $10^{-4}\,\mathrm{s^{-1}}$ in the simulation set spanning these ranges of Rossby numbers. However, we remind that Rossby similarity means that for a given pair of $\{\mathrm{Ro}_0, \mathrm{Ro}_h\}$ and $\{z_0, h\}$ one has many (infinite) combinations of $\{|G|, f\}$ which give the same $|G|/f$ and thus the same dimensionless profile shapes of velocity, i.e. speed and direction as a function of dimensionless height $zf/|G|$. At any rate, the simulations cover ranges of (exceeding): ABL depths of 200–2000 m; roughness lengths from water's roughness

(0.1 mm) up to 2.5 m; and $|G|$ from 5–50 m s$^{-1}$.

### 3.1.2 Shear and veer over neutral ABLs simulated over entire range of Rossby numbers found in nature

First we check that the RANS simulations confirm the shear-veer relations developed earlier; we expect this to be, since there are no extra terms in the simulated Navier-Stokes equations compared to (7). Figure 2 displays both sides of (9) and (13) respectively, for four cases representing somewhat common real-world conditions, for heights between 50–200 m. From Fig. 2

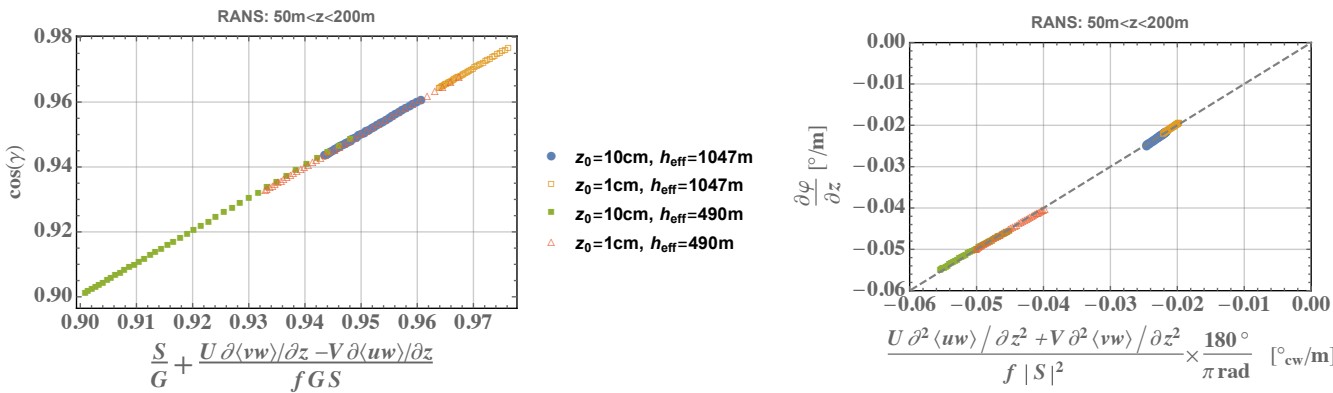

**Figure 2.** Demonstration of 1D RANS solver results, conforming to eq. 13 (left) and eq. 9 (right). Dashed line represents 1:1 prediction; simulated ABL depth $h_{\mathrm{eff}}$ calculated from (35).

one can see that the Ellipsys1D solutions conform to equations (9) and (13) derived earlier.

Towards investigating the behavior of veer (and shear) in terms of Rossby numbers – which is facilitated by RANS, but is quite difficult to accomplish with measurements – we turn our attention to the variation in veer as a function of surface roughness. Admitting that we are using one-dimensional simulations over a homogeneous surface, we now consider the directional change across typical turbine rotor heights, i.e. $\Delta\varphi$ from $z = 50\,\mathrm{m}$ to 150 m. Figure 3 displays $\Delta\varphi|_{50\,\mathrm{m}}^{150\,\mathrm{m}}$ plotted over different

roughnesses for the two ABL depths represented in the cases shown of the previous figure, namely 490 m and 1047 m. From



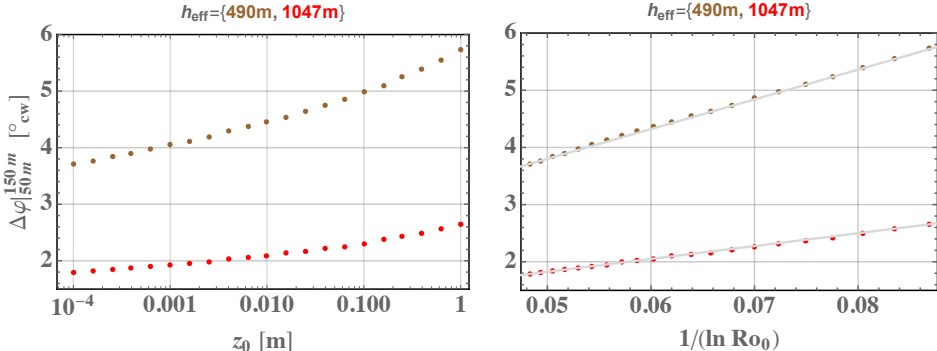

**Figure 3.** Roughness dependence of turning (in degrees, clockwise) seen for two representative ABL depths, from 1D RANS simulations over a range of roughness lengths plotted directly against $z_0$ (left) and alternately versus $1/\ln(\text{Ro}_0)$ (right). Lines in right-hand plot indicate linear trend.

the right-hand plot in Fig. 3 one can see that $\Delta\varphi$ is roughly proportional to $1/\ln(\text{Ro}_0)$, as expected from the $S/|G|$ contribution to veer considering (30) and (32).

Looking back on (33), we may also expect a $\text{Ro}_h$ dependence in the veer, due to the stress gradient contributions. Figure 4 shows veer across three different rotor extents ($z =$ 50–100 m, 50–150 m, and 100–200 m), over a wide range of effective ABL depth $h$ and associated Rossby number $\text{Ro}_h$, for a commonly found roughness over land (1.6 cm). In Fig. 4 results are shown

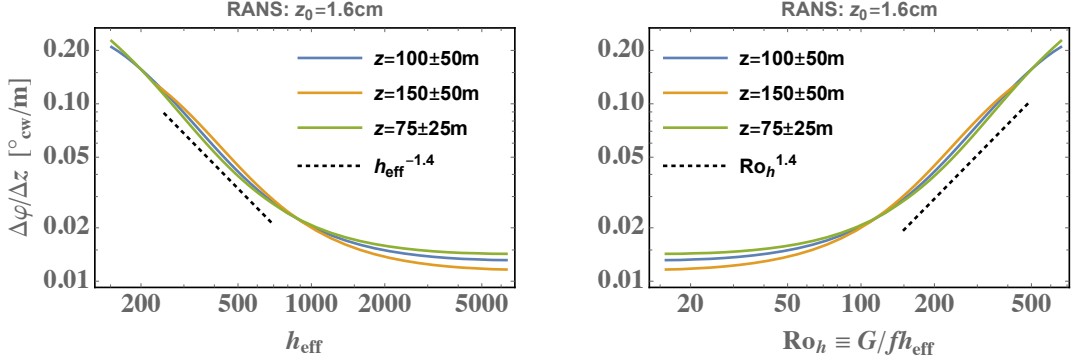

**Figure 4.** Influence of ABL depth and associated Rossby number on veer (clockwise) for different turbine rotor spans.


only for one roughness, because the curves of veer versus ABL depth and $\text{Ro}_h$ look nearly identical when using any other $z_0$ (or $\text{Ro}_0$) value, such as water roughnesses less than 0.3 mm. In other words, the sensitivity of veer to $h$ is essentially independent of $z_0$, if one varies these separately from case to case as in our numerical simulations. Looking at these results, we note a behavior that is not inconsistent with the estimates for stress-gradient contributions following (33): the veer is proportional to

$\text{Ro}_h^{1.4}$ (or $h^{-1.4}$) over a range of ABL depths routinely observed in reality ($h \sim$ 200–800 m), though the dependence softens to be linear in $\text{Ro}_h$ (or $1/h$) for depths approaching $h \sim$1 km, which are also commonly observed in nature. For yet deeper ABLs





which are more rarely encountered, the height dependence vanishes; this can be intuitively interpreted, as $\Delta z/h$ becomes so small that less directional change is found for a given $\Delta z$ when $h$ is increased further. The veer and its $h$-dependence is seen to be basically independent of height for these 50–100 m vertical spans: at the heights of interest for wind energy shown, the lines collapse onto one another. To compare with Fig. 3, multiplying the veers in Fig. 4 by $\Delta z = 100\,\mathrm{m}$ for the blue and gold curves, we can also see that for a realistic range of ABL depths and roughnesses, the effect of $h$ is stronger than that of $z_0$: across all $\mathrm{Ro}_0$ a variation in $\Delta\varphi|_{50\,\mathrm{m}}^{150\,\mathrm{m}}$ of only several degrees is seen, whereas across the common range of $\mathrm{Ro}_h$ a variation of more than $15°$ is shown.

Now that we have seen in Figs. 3–4 how the veer (or simply the turning $\Delta\varphi$ for typical rotor $\Delta z$) depends on $z_0$ and $h$, presumably due to the $S/|G|$ (shear) and stress-gradient contributions respectively, it is prudent to examine the relative sizes of each of these contributions – particularly because RANS affords us this opportunity. One can cleanly separate these contributions by examining the variation of $\cos\gamma$, as indicated by (12) and (13). Accordingly, Fig. 5 presents the two contributions to the dimensionless veer $\partial\cos\gamma/\partial z$ derived in (12), for the four over-land cases shown in Fig. 2 as well as an over sea case with the same ABL depth as two of the land cases.

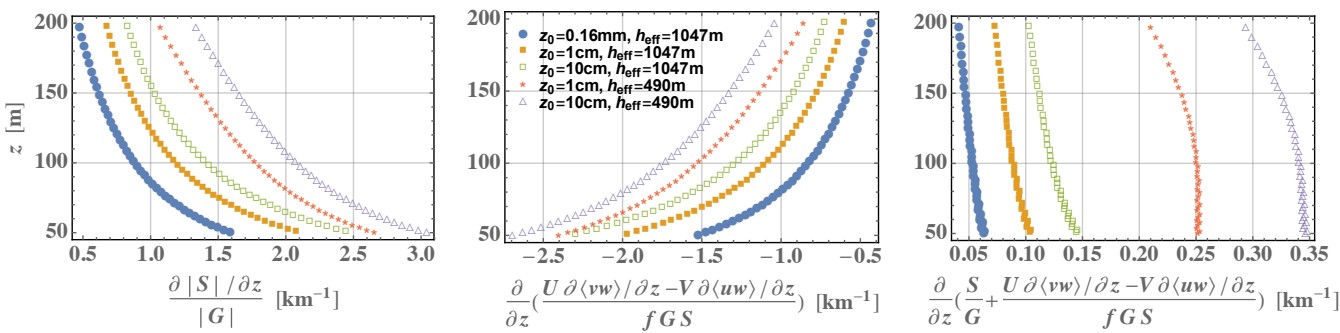

**Figure 5.** Profiles of contributions to $d\cos\gamma/dz$ in (12) due to shear (left), stress gradients with Coriolis (center), and their sum (right). Five RANS simulations shown (two roughnesses and two ABL depths over land, one over sea) over typical turbine rotor heights; the listed $z_0$ and $h$ correspond to Rossby numbers using $G = 10\,\mathrm{m\,s^{-1}}$ and $f = 10^{-4}\,\mathrm{s^{-1}}$.

One can note from Fig. 5 that the shear and stress-gradient/Coriolis contributions largely offset each other, with each being an order of magnitude larger than their sum, which is equal to the dimensionless veer $\partial\cos\gamma/\partial z$. The vertical profiles of 'pointwise' veer shown in the figure, which were calculated using 3rd-order finite difference, indicate that in neutral conditions the veer is smaller offshore compared to on land. Further, one sees the combined effect of the behaviors noted from the previous two figures: shallower ABLs have larger veer, as do ABLs over rougher surfaces, with $\mathrm{Ro}_0(z_0)$ having a smaller impact than $\mathrm{Ro}_h(h)$.

This can be put into a more practical context by considering the variation of shear and veer together across the range of Rossby numbers found in atmospheric flows. Figure 6 displays turning versus shear exponent, with each calculated across $\Delta z$ from 50–150 m. The figure shows three plots of $\{\alpha, \Delta\varphi\}$: one for a range of $\mathrm{Ro}_h$ equivalent to $h$ ranging from 490 to 1047 m, over two different $z_0$ (land and sea); one for a range of $\mathrm{Ro}_0$ equivalent to $z_0$ values varying from 0.016–25 cm, for two different





400 ABL depths $h$ (which bracket the range of $h$ in the left-hand plot); and one over the entire atmospheric range of both $\mathrm{Ro}_0$ and $\mathrm{Ro}_h$.

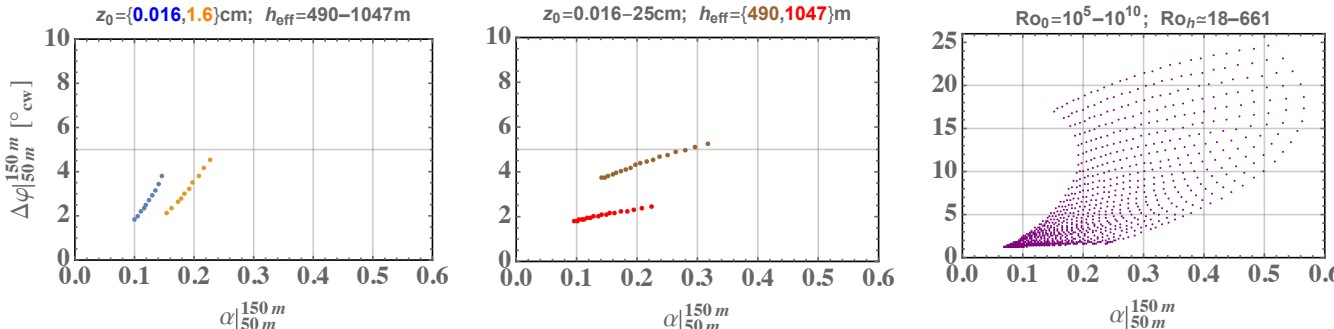

**Figure 6.** Turning (bulk veer) in degrees clockwise versus shear exponent calculated from 50–150 m, over ranges of ABL depth and surface roughness; each point represents one RANS solution. Left: using $G = 10\,\mathrm{m\,s^{-1}}$ and $f = 10^{-4}\,\mathrm{s^{-1}}$, over range of ABL depths spanning the values used in Figs. 2–5 for water and typical land roughness. Center: again with $G = 10\,\mathrm{m\,s^{-1}}$ and $f = 10^{-4}\,\mathrm{s^{-1}}$, over range of $z_0$ spanning those used in Figs. 2–5, for two ABL (typical) depths used in previous figures. Right: over wider range of $\{\mathrm{Ro}_0, \mathrm{Ro}_h\}$ spanning that found in nature; note larger vertical axis scale.

From the left and center panels of Fig. 6, it becomes evident that $\mathrm{Ro}_h$ affects $\Delta\varphi$ more than $\alpha$ for typical rotor extents; opposite of this, $\mathrm{Ro}_0$ affects the shear more than the veer. Further, for the relatively representative set of (common) cases shown in the center and left-hand plots in Fig. 6, we notice much less variation in $\{\alpha, \Delta\varphi\}$ compared to the entire parameter space displayed in the right-hand plot; as we will see in the next sub-section, the right-hand plot is more in line with observations, 405 despite the RANS solutions representing nominally neutral conditions[10] over uniform surfaces with neglect of shear-stress misalignment and baroclinity.

### 3.2 Results from measurements in different wind regimes and sites

After examining the behavior of neutral-ABL dependencies for shear and veer above from simulations, now we consider the 410 behavior of each in the real world from measurements at different sites, which includes e.g. the affects of stability. The datasets are the same analyzed by Kelly et al. (2014a), which showed shear exponent statistics for these locations, except a longer record of Høvsøre data was used for the current study (10 years, from 2005–2015). These are: the aforementioned Høvsøre site, from 60–160 m height for both homogeneous land and sea sectors; the partly forested but flat Østerild site (Hansen et al., 2014) for two virtual rotor spans, from 45–140 m and 80–200 m over one year; the Dutch research site Cabauw (Beljaars and Bosveld,

---

[10]One could argue that our RANS solutions can also be interpreted to include stable conditions, since the lengthscale-limited $k$-$\varepsilon$ turbulence model can have its maximum mixing length $\ell_{\max}$ rewritten using the Blackadar (1962) mixing-length formulation such that $\ell_{\max,\mathrm{eff}}^{-1} = \ell_{\max}^{-1}$ plus a stability contribution, as shown in van der Laan et al. (2020). However, such interpretation employs M-O theory along with the Blackadar-type form to 'combine' a surface-layer scale $\ell_{\mathrm{ASL}} \propto z$ with $\ell_{\max}$; here we choose to keep our analysis as general as possible — avoiding particular ASL forms or assumptions, and models for turbulence length scale.



1997), from 80–200 m height for two years; and one year from a commercial site dubbed 'MR' which sits on a ridge over a mostly forested ($> \sim 3/4$) area but dominated by hills having elevation differences up to $\sim$200 m within 10 km distance, using anemometers at 40–136 m height.[11]

We investigate the statistical behavior of veer with shear exponent as well, not only to see their interdependent behavior, but also towards providing useful relations for their variability and practical prediction of veer from typical wind energy measurement campaigns.

### 3.2.1 Shear exponent

Here we briefly explore the connection between probability distribution functions (PDFs) of stability and shear exponent. The shear distribution $f(\alpha)$ can be connected to $f(L^{-1})$ in the surface layer during stable conditions, but there is not necessarily a one-to-one (unique) mapping between the two (Kelly et al., 2014a). As seen in (4) and (5), $\alpha$ tends to correlate with stability ($1/L$) and particularly buoyant destruction ($-B$) during stable conditions, when turbulent transport is negligible. This is shown in Fig. 7, which displays the joint probability density of $\alpha|_{60\,\text{m}}^{160\,\text{m}}$ and $L^{-1}$ calculated in the ASL at $z = 10$ m from the homogeneous land sectors at the Danish national test station of Høvsøre (Peña et al., 2016) from 10-minute averages over a 10-year period.

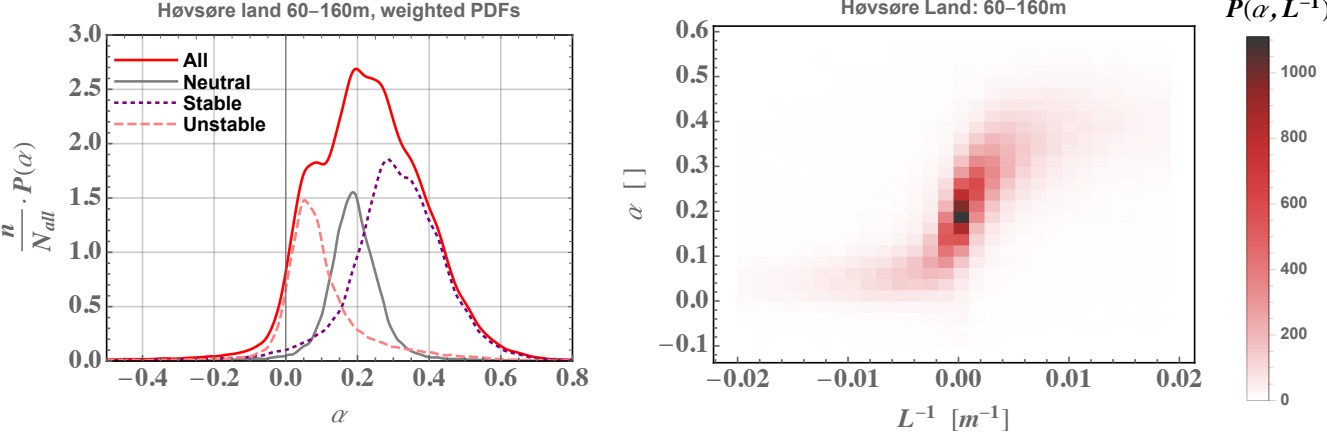

**Figure 7.** Left: PDF of shear exponent $\alpha$, weighted by frequency of occurence for different stability conditions; neutral defined by $|L^{-1}| < 0.001\text{m}^{-1}$, stable has $L^{-1} > 0.001\text{m}^{-1}$, and unstable has $L^{-1} < -0.001\text{m}^{-1}$. Right: joint probability distribution of shear exponent calculated from 60–160 m height and inverse Obukhov length (surface-layer stability) at $z = 10$ m from sonic anemometers over the homogeneous land sectors at Høvsøre. Measurements span one decade, starting 2005.

From Fig. 7 one sees the most likely values of $\{\alpha, L^{-1}\}$ follow a curve which resembles the nondimensional M-O shear function $\Phi_m(z/L)$. This is not surprising considering (2) and (5), though we remind that the upper-level height (160 m) used

---

[11]The details and location of the site 'MR' cannot be shared publicly, due to their proprietary nature (see also Kelly et al., 2014a). The site is located near the border between New York state (USA) and Canada, in a moderately hilly region.



in the $\alpha$ calculation is above the surface layer. The left-hand panel of Fig. 7 also shows the distribution of $\alpha$ for neutral ($|L^{-1}| < 0.001\,\mathrm{m}^{-1}$), stable ($L^{-1} > 0.001\,\mathrm{m}^{-1}$), and unstable ($L^{-1} < -0.001\,\mathrm{m}^{-1}$) flow regimes, weighted by frequency of occurrence to show the relative contributions to the overall distribution. The threshold of $\pm 0.001\,\mathrm{m}^{-1}$ for $L^{-1}$ is a sensible choice because then $z/|L| \ll 1$ (consistent with neutral conditions) in the surface layer, which is generally taken to have a
thickness of $100\,\mathrm{m}$ or less (roughly $h/10$, also recalling that M-O theory's applicability diminishes with height above the surface-layer). Even at this relatively flat and uniform site, negative shear happens in both stable and unstable conditions, though moreso in unstable and yet less often in neutral conditions; overall, $\alpha < 0$ occurs less than 5% of the time over 60–160 m here, and 8–9% from anemometers at 100–160 m heights (not shown). We also note that while the 'ideal' Høvsøre land (eastern) sectors have conditions split somewhat evenly between the three stability regimes, other sites can differ (Kelly and
Gryning, 2010).

### 3.2.2 Veer

Along with distributions of $\alpha$, measured veer distributions are shown in Fig. 8 for both land and sea conditions at Høvsøre, i.e. from the homogeneous offshore/open-fetch ($240° < \varphi < 300°$) and over-land ($60° < \varphi < 120°$) directions. Shear and veer are shown calculated over height spans of 60–160 m as well as 100–160 m in the figure, which is provided to show the statistical
and behavioral differences between shear and veer.

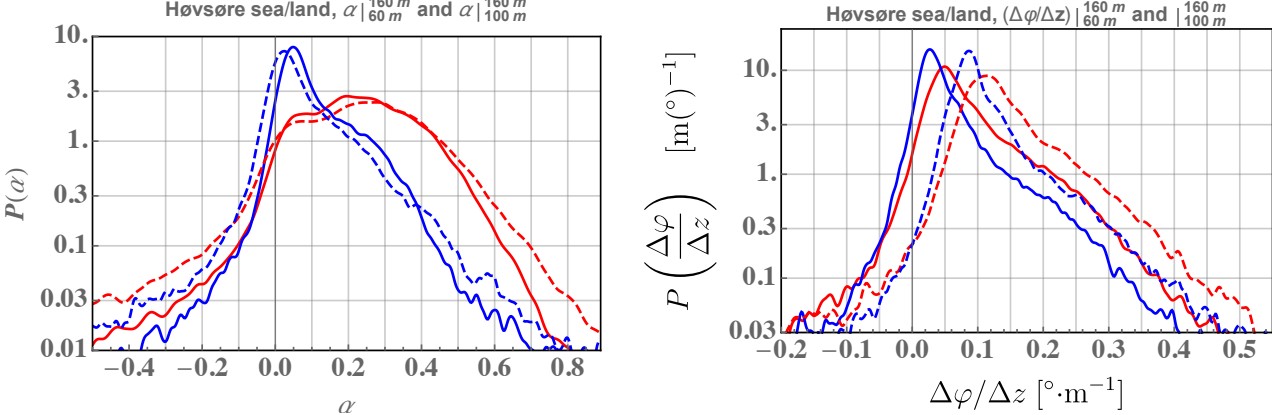

**Figure 8.** Distributions of shear exponent (left) and corresponding veer (right) at Høvsore between 60–160 m, from the homogeneous eastern land sectors (red) and from the sea sectors to the west (blue). Solid lines are for measurements spanning 60–160 m; dashed for those spanning 100–160 m.

From the two plots one can see that the most common $\alpha$ and $\Delta\varphi/\Delta z$, i.e. the portions of $P(\alpha)$ and $P(\Delta\varphi/\Delta z)$ with respective probabilities within an order of magnitude of the peak values, both systematically differ when using higher measurements at 100–160 m compared with 60–160 m heights; however, the shift in the commonest $\alpha$ is significantly smaller than the analogous shift in $\Delta\varphi/\Delta z$ between these two height ranges. This happens over both land and sea, though both $\alpha$ and $\Delta\varphi/\Delta z$
vary with height more for the offshore flow than for the homogeneous land directions. The change of $\langle\alpha\rangle$ from 60–160 m to



100–160 m is less than $+5\%$ over land and $< +30\%$ is seen over sea, while the mean veer $\langle \Delta\varphi/\Delta z \rangle$ is seen to increase by factors of $\sim 5/3$ and 2 over land and sea, respectively.

There are several other notable differences between the shear and veer statistics shown in Fig. 8. The peak portion of $P(\alpha)$ is significantly wider over land compared to offshore (with larger $\sigma_\alpha$ over the rougher surface), while the shape around the

$P(\Delta\varphi/\Delta z)$ peak does not differ significantly from land to sea here. Further, the (logarithmic) slope of $P(\Delta\varphi/\Delta z)$ versus $\Delta\varphi/\Delta z$ for veer larger than the PDF peak is basically the same regardless of height or surface conditions; this and the land-sea difference between $P(\alpha)$ are consistent with the earlier RANS results, where $z_0$ primarily affects $\alpha$, while $\Delta\varphi/\Delta z$ is impacted more by ABL depth. $P(\alpha)$ also has wider 'tails' (extremes) higher from the ground on both sides, including negative shear due to low-level jets (such as that due to the capping-inversion when $h \sim 200$ m), whereas the veer simply becomes larger due

to such jets in shallow ABLs, as jets and the environment associated with the capping inversion simply causes more turning, and not a reversal. The negative veer occurs due to nonstationary processes like passing fronts (e.g. Clark, 2013), as well as baroclinity and motions associated with it (Arya, 1978; Foster and Levy, 1998; Floors et al., 2015). Comparing the solid and dashed lines in Fig. 8, one sees that the highest veers $\Delta\varphi/\Delta z$ are larger for the 100–160 m measurements than those from 60–160 m; this is again due to more impact of the ABL-capping inversion and associated jet with turning.

As with shear, stability affects veer, with stable conditions expected to lead to higher veer due to its damping effect on vertical fluxes (suppressing vertical 'communication' of flow information). Following the plots shown in Fig. 7 for the shear exponent $\alpha$, Fig. 9 displays the effect of stability on veer for the Høvsøre land sectors. The figure shows $P(\Delta\varphi/\Delta z)$ for neutral ($|L^{-1}| < 0.001 \mathrm{m}^{-1}$), stable ($L^{-1} > 0.001 \mathrm{m}^{-1}$), and unstable ($L^{-1} < -0.001 \mathrm{m}^{-1}$) flow regimes, weighted by frequency of occurrence (indicating relative contributions to the full PDF), as well as the joint distribution of stability and $\Delta\varphi/\Delta z$.

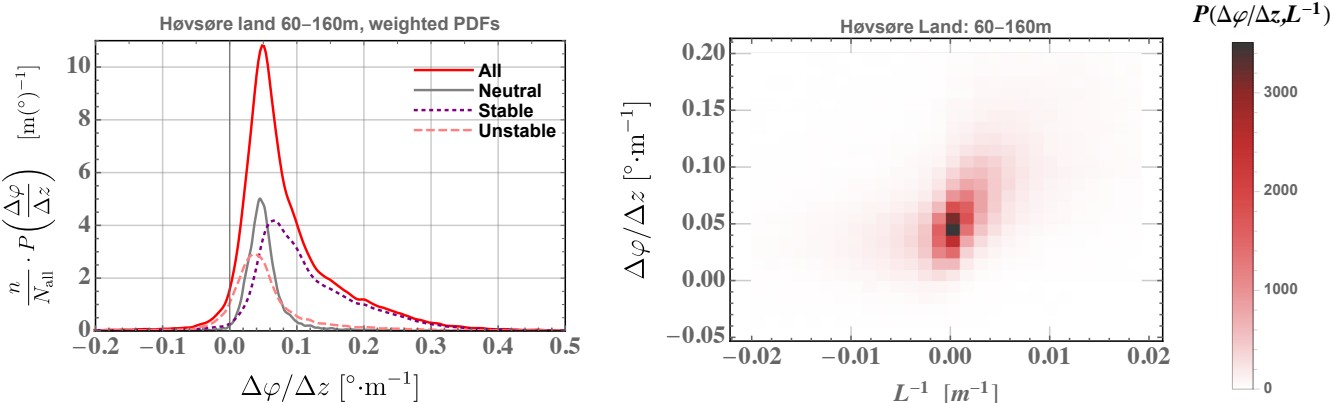

**Figure 9.** Left: PDF of veer $\Delta\varphi/\Delta z$, weighted by frequency of occurence for different stability conditions; neutral has $|L^{-1}| < 0.001 \mathrm{m}^{-1}$, stable has $L^{-1} > 0.001 \mathrm{m}^{-1}$, and unstable has $L^{-1} < -0.001 \mathrm{m}^{-1}$. Right: joint probability distribution of veer calculated from 60–160 m height and inverse Obukhov length (surface-layer stability) at $z = 10$ m from sonic anemometers over the homogeneous land sectors at Høvsøre. Measurements span one decade, starting 2005.





From Fig. 9 one sees that in comparison with $P(\alpha)$ shown in Fig. 7, the *peaks* of veer distributions $P(\Delta\varphi/\Delta z)$ do not depend so much on stability. However, as with the shear distribution, $\Delta\varphi/\Delta z$ also has its largest values dominated by stable conditions; this makes sense considering that stability tends to maintain vertical gradients by limiting vertical fluxes. Unlike the results shown for the RANS simulations or predicted by theory, negative veer occurs as in Fig. 8 and described thereunder; one can see in Fig. 9 that it basically happens during non-neutral conditions, which tend to occur at lower wind speeds, and

is dominated by unstable conditions. Looking at the joint distribution $P(\alpha, \Delta\varphi/\Delta z)$ one sees that for the most common veer values ($0 \lesssim \Delta\varphi/\Delta z \lesssim 0.1°/\mathrm{m}$), which tend to occur around neutral conditions, there is a mild stability dependence; however for less neutral conditions there is little correlation between veer and stability, aside from higher veer simply being observed more often in stable conditions.

To show the behavior of veer across different locations, Fig. 10 displays the PDFs of veer from a number of sites, all of which have similar $\Delta z$ and cover typical turbine rotor extents. From Fig. 10 we see that for veer magnitudes exceeding the

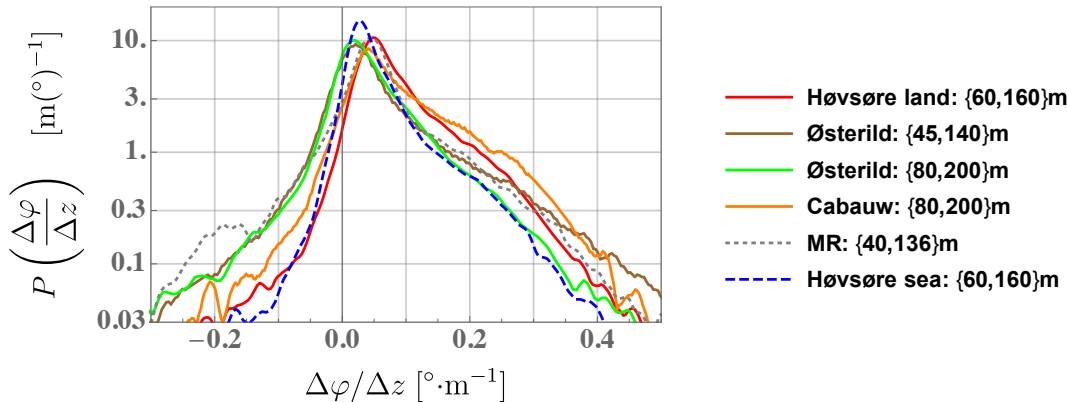

**Figure 10.** Probability density function (distribution) of veer, $P(\Delta\varphi/\Delta z)$, for all conditions at the various sites/cases considered.


most commonly observed values (which tend to occur in stable conditions, as shown in Fig. 9 for the Høvsøre case above), the distributions behave similarly across locations; in particular the "slope" of the semi-log plot for veer exceeding the PDF peaks is roughly constant for $\Delta\varphi/\Delta z \gtrsim 0.2° \, \mathrm{m}^{-1}$ in each case. These slopes correspond to (conditional) PDFs for the largest veer of the form

$$P\left(\frac{\Delta\varphi}{\Delta z}\middle|\frac{\Delta\varphi}{\Delta z} > \mathrm{mode}\left\{\frac{\Delta\varphi}{\Delta z}\right\}\right) \propto \exp\left[\frac{-\Delta\varphi/\Delta z}{\Upsilon_{\mathrm{veer}}}\right], \tag{36}$$

where the charcteristic veer scale defined by $\Upsilon_{\mathrm{veer}}^{-1} \equiv \partial[\ln P(\Delta\varphi/\Delta z)]/\partial(\Delta\varphi/\Delta z)$ ranges from roughly 0.07 to 0.11 $°\cdot\mathrm{m}^{-1}$. The lowest $\Upsilon_{\mathrm{veer}}$ corresponds the offshore Høvsøre case, while the highest $\Upsilon_{\mathrm{veer}}$ matches the Østerild case from 45–140m. We expect larger $\Upsilon_{\mathrm{veer}}$ to correspond to occurences of higher $1/L$, i.e. a larger width $\sigma_+$ of the stable-side distribution $P(1/L)$ following Kelly and Gryning (2010); essentially the large-veer PDF in (36) is conditional on stable conditions, i.e. we could

express it as $P\left(\Delta\varphi/\Delta z | L^{-1} > 0\right) \propto \exp\left[-(\Delta\varphi/\Delta z)/\Upsilon_{\mathrm{veer}}\right]$. The dominance of stable conditions reported by Peña (2019)



for $z \gtrsim 100$m at Østerild is consistent with this, though the data from $z =80$–200m (green line in Fig. 10) with smaller apparent $\Upsilon_{\mathrm{veer}}$ might appear to not be, considering the increasingly stable conditions higher up at this site; but looking at the Østerild curves in the figure we see that for higher veer $\Delta\varphi/\Delta z \gtrsim 0.4°\cdot$m$^{-1}$, there is consistency: the two largest $\Upsilon_{\mathrm{veer}}$ occur for $z =80$–200m and $z =45$–140m, respectively. Future work needs to be done to explore this, since we lack air-sea temperature differences (or water-air heat flux) for the Høvsøre offshore case and stability information for the MR site, while stability effects above forests tend to be diminished and are difficult to interpret due to turbulent transport through the treetops (e.g. Sogachev and Kelly, 2016).

One also sees the peaks of $P(\Delta\varphi/\Delta z)$ in Fig. 10 are at smaller $\Delta\varphi/\Delta z$ for the forest-dominated Østerild cases, with the peak of the offshore Høvsøre veer distribution falling between these and the $\Delta\varphi/\Delta z$ corresponding to the the land cases of Høvsøre, Cabauw, and 'MR'. We remind that the most commonly-found veer values are generally dominated by neutral conditions (or modestly stable for the exceptional Østerild site above 100m), and point out that the mode of $\Delta\varphi/\Delta z$ is essentially the same (0.005–0.006$° \cdot$m$^{-1}$) for the land cases that are not dominated by forest. Further considering the RANS simulation results from Fig. 6 discussed earlier, the mode of $\Delta\varphi/\Delta z$ being smaller for Høvsøre offshore than for the land cases (of Høvsøre, Cabauw, and 'MR') can be explained by the smaller ABL depths most commonly observed offshore compared to onshore; this is consistent with the ABL depth distributions aggregated and reported by Liu and Liang (2010). The mode of $\Delta\varphi/\Delta z$ found at the inhomogeneous forest-dominated site Østerild are more strongly affected by the tree-enhanced mixing (which reduces the veer magnitudes) and to a lesser extent by shallower ABLs due to the coastline 5–20km upwind in some directions.

The dependence of veer on wind speed at the sites considered is shown in Fig. 11, which displays the joint distribution of veer and 10-minute mean wind speeds, $P(\Delta\varphi/\Delta z, U)$. Along with the joint distribution, the mean veer conditioned on wind speed, $\langle\Delta\varphi/\Delta z\rangle|U$, is displayed.

From Fig. 11 one can see results consistent with the effects of stability discussed earlier and evoked by Fig. 9: at higher speeds neutral conditions dominate, giving decreased mean veer. This is more pronounced for the onshore cases (though there is still a reduction of nearly 40% going from 12 to 24 m s$^{-1}$ for the offshore case), because sea-air heat fluxes and associated $1/L$ magnitudes tend to be relatively smaller due to water's large heat capacity (e.g. Cronin et al., 2019). It is notable that for the representative wind turbine rotor heights considered, the veer tends to be largest for wind speeds below typical turbine rated speeds, especially over land; this can have consequences on both the power output and effective power curve for pre-construction AEP estimates, as well as loads.

Further, a narrower range of veer with increasing wind speed is seen in Fig. 11, regardless of surface properties; such narrowing is impacted by stability, but also occurs in neutral conditions. The variability of veer with mean wind speed is presented in Fig. 12, which displays the standard deviation of veer conditioned on mean wind speed for the sites/cases considered. It also adds a line to show the overland Høvsøre case filtered for neutral conditions.

Consistent with the joint-PDFs $P(\Delta\varphi/\Delta z, U)$ in Fig. 11, from the semi-logarithmic plot of standard deviation of veer conditioned on mean wind speed in Fig. 12 we can see that the variation in veer decreases with wind speed, and moreso over land than water. It is also seen that for the onshore Høvsøre case $\sigma_{(\Delta\varphi/\Delta z)|U}$ is smaller in neutral conditions compared to over





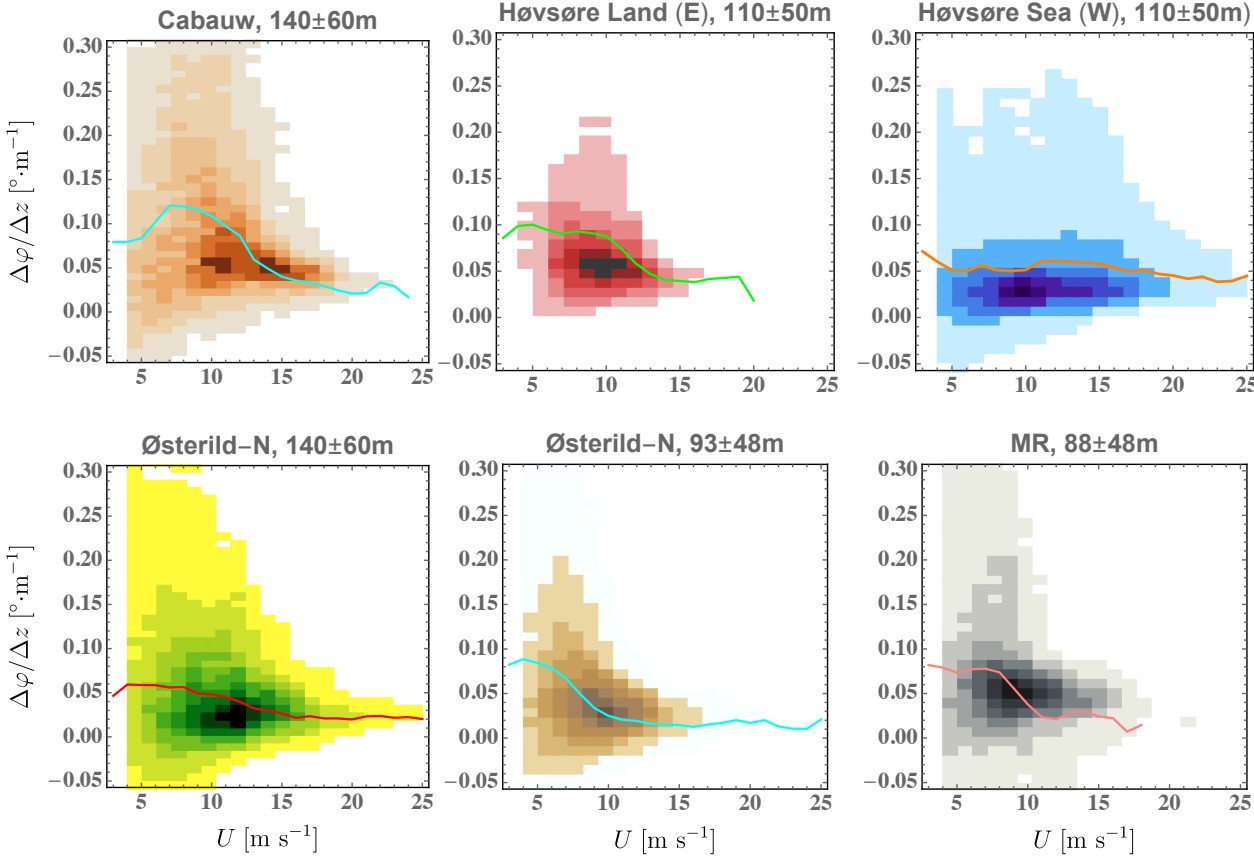

**Figure 11.** Joint distribution of veer and wind speed at sites considered. Solid line shows $\langle \Delta\varphi/\Delta z \rangle | U$, calculated using $1\,\mathrm{m\,s^{-1}}$ bins; lightest shades are 2% as likely as darkest color in each plot.

all stabilities, with the two values converging at higher speeds due to the increasingly neutral conditions. For each site having a standard deviation of veer over all speeds $\sigma_{\Delta\varphi/\Delta z}$ and mean wind speed $\langle U \rangle$, the rms veer conditioned on wind speed roughly follows the empirical form

$$\sigma_{(\Delta\varphi/\Delta z)|U} = \left[ \left\langle \left( \frac{\Delta\varphi}{\Delta z} \right)^2 \middle| U \right\rangle \right]^{1/2} \approx \sigma_{\Delta\varphi/\Delta z} \exp\left[ \frac{-U}{\langle U \rangle} \right] \tag{37}$$

up to about $12\,\mathrm{m\,s^{-1}}$ over land, and to higher speeds offshore. A more complicated speed-dependent variability in veer is seen for the MR case, with higher $\sigma_{(\Delta\varphi/\Delta z)|U}$ at speeds above $15\,\mathrm{m\,s^{-1}}$ caused (presumably) by hill-induced turning. This has two consequences worth mentioning: first, that turbines at a site such as 'MR' can experience persistent veer above rated speed, potentially increasing loads and/or reducing power below rated; secondly, such speed-dependent behavior is likely difficult to capture with standard single RANS simulations, demanding more detailed treatment to handle the Reynolds-number

dependence despite the lack of stability effects at such speeds.

WIND
ENERGY
SCIENCE
DISCUSSIONS

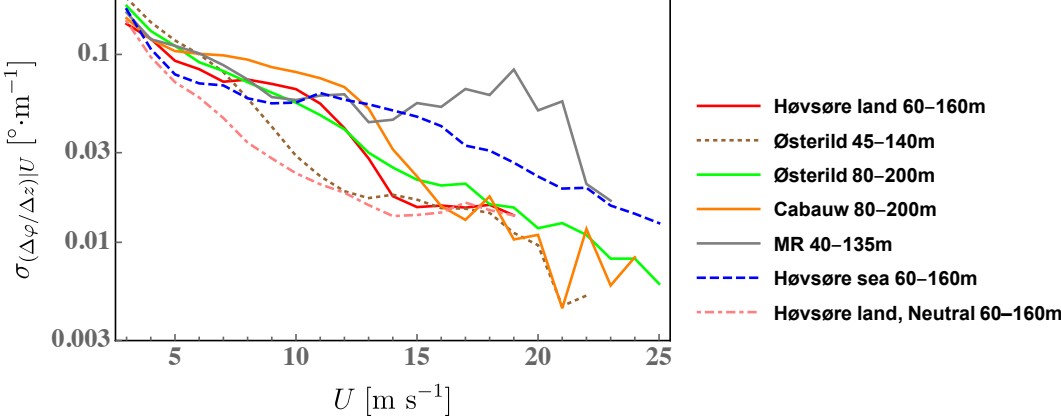

**Figure 12.** Measured standard deviation of veer conditioned on wind speed, again using $1\,\mathrm{m\,s^{-1}}$ bins, for the sites considered. Neutral conditions at Høvsøre defined as in earlier figures, i.e. $|L^{-1}| < 0.001\,\mathrm{m^{-1}}$.

### 3.3 Relating veer to shear in application

One of the aims of this work is to relate veer to shear (or shear exponent), as with the expressions developed in Sect. 2.3–2.4. Here we present joint observations of shear exponent and veer, and following these, give practical simplified forms based on the equations derived earlier in sections 2.3–2.4.

Following the previous subsection, we first consider the joint behavior of $\Delta\varphi/\Delta z$ and $\alpha$ with wind speed and stability, for the 'simple' onshore Høvsøre case having homogeneous upwind conditions. Figure 13 shows the observed joint distribution $P(\Delta\varphi/\Delta z, \alpha)$ in neutral conditions, over typical turbine operation speeds ($4$–$25\,\mathrm{m\,s^{-1}}$) and separately over different speed ranges ($4$–$8$, $8$–$12$, $12$–$16$, and $16$–$25\,\mathrm{m\,s^{-1}}$); counts are used instead of PDF per wind speed range, to show relative frequencies of occurence.

From Fig. 13 one can notice that in neutral conditions there does not appear to be significant variation in the joint shear-veer behavior with $U$, with a bit more variability at the lowest speeds and smaller values of both $\Delta\varphi/\Delta z$ and $\alpha$ for $U > 16\,\mathrm{m\,s^{-1}}$; this is consistent with Figs. 7, 9, 11, and 12. The larger spread at lower speeds for neutral conditions is attributed to the larger relative effect of non-stationarity and particularly sampling uncertainty; per the latter the integral time scale increases roughly as $U^{-1}$ (Wyngaard, 2010) so fewer integral time scales are 'sampled' per each 10-minute period. This is also evident

considering the previous plot of $\sigma_{(\Delta\varphi/\Delta z)|U}$ versus $U$ in Fig. 12, where one sees $\sigma_{(\Delta\varphi/\Delta z)|U}$ increasing with diminishing wind speed during both neutral and all conditions for the Høvsøre land case, but where stability effects cause larger veer variability up to speeds of about $15\,\mathrm{m\,s^{-1}}$. Also, the overall jPDF $P(\Delta\varphi/\Delta z, \alpha)$ appears similar to that in the most common speed range ($8$–$12\,\mathrm{m\,s^{-1}}$). Aside from nonstationarity and sampling effects one does not expect much speed dependence in neutral conditions, considering the $\alpha$-related part of (14)–(16) behaves as $|S|/G$, which following (32) has a weak $|S|$-dependence

through $(\ln\mathrm{Ro}_0 - A)^{-1}$; the RANS results also confirm this. We note a joint trend between $\alpha$ and $\Delta\varphi/\Delta z$, but also see a


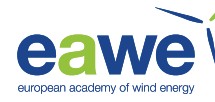
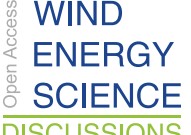

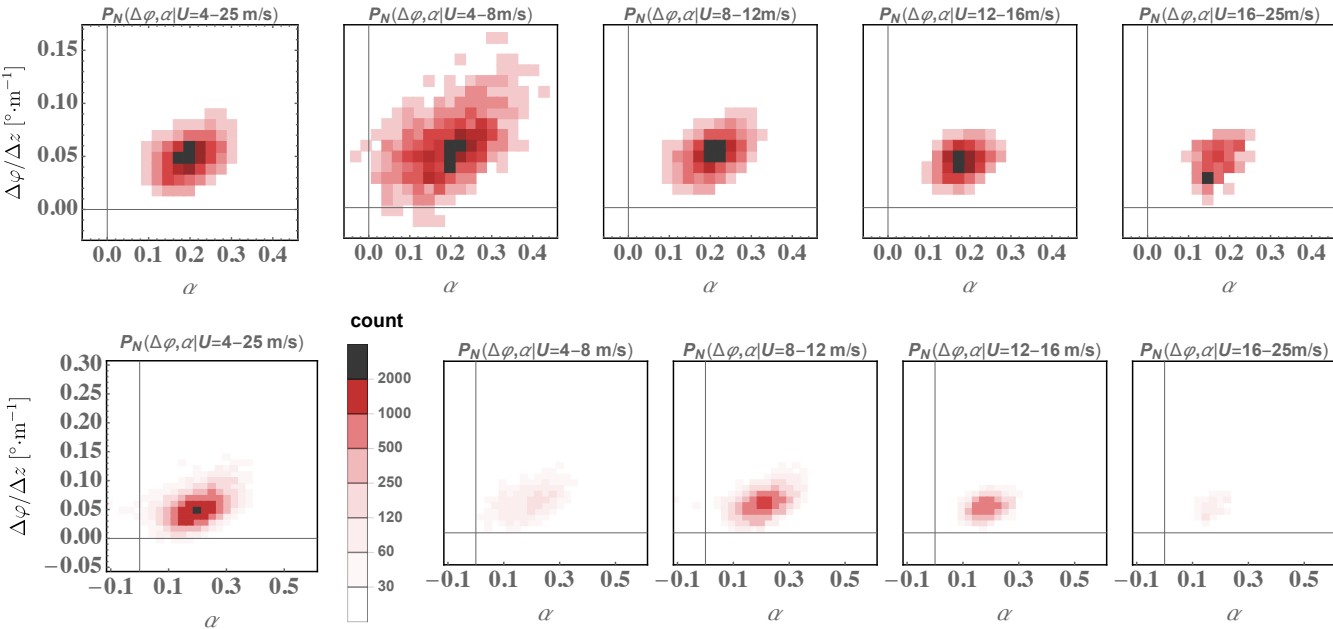

**Figure 13.** Top: joint distribution of veer and shear exponent observed over 10 years from 60–160 m for the Høvsøre land sectors in neutral conditions, in different speed ranges; axes zoomed in to show detail, and occurence rate normalized per wind speed range (each plot has a different color scale, showing occurence rate in increments of 1/10, with lightest representing 10% as likely as darkest shade). Bottom: the same joint distributions shown with unscaled rate of occurrence (number of counts per $\{\alpha, \Delta\varphi/\Delta z\}$ bin); axis ranges are chosen to compare with later figures.

spread around the most common shear exponent and veer values due to variations in ABL depth, stress gradient and curvature, and top-down stability (capping-inversion strength, see e.g. Kelly et al., 2019a), in addition to nonstationarity.

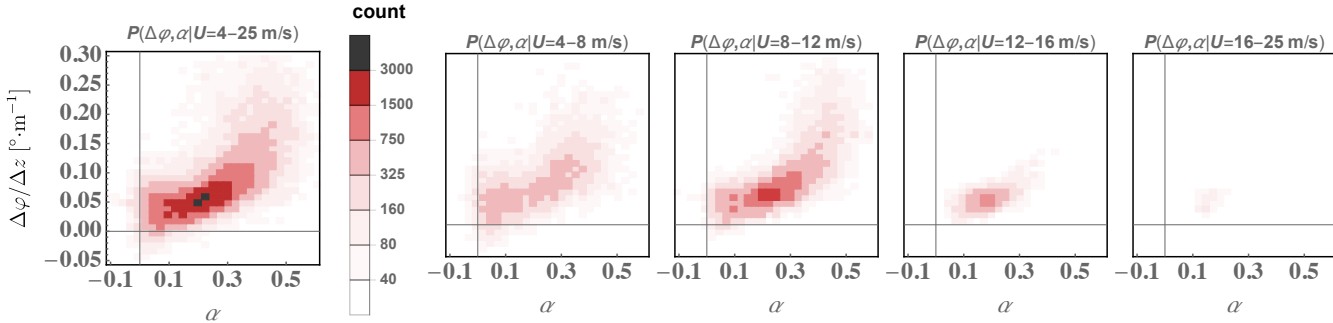

**Figure 14.** Joint distribution of veer and shear exponent for different speed ranges from Høvsøre land sectors, but for all stability conditions; plots are analogous to those in bottom of Fig. 13. All plots use same color scale; color bar denotes count.





Figure 14 shows joint $\alpha$-veer distributions like Fig. 13, but over all conditions, i.e., not limited to neutral stratification. One notices immediately the more frequent occurence of higher veer and shear, as well as negative $\alpha$ and $\Delta\varphi/\Delta z$. Further, in addition to a wider range of shear and veer compared to neutral conditions, in Fig. 14 one can see there is also a sharper increase in $\Delta\varphi/\Delta z$ with $\alpha$ for larger $\alpha$, due to stable conditions. One can see that at the most common (8–12 m s$^{-1}$) and lower wind speeds, which occur in the range below rated speed for typical turbines, there is a significant increase in $\Delta\varphi/\Delta z$ with $\alpha$ in the more stable conditions where $\alpha \gtrsim 0.3$; this higher 'slope' of $\Delta\varphi/\Delta z$ vs. $\alpha$ is likely enhanced by the shallower ABLs which generally occur along with stable surface-layer conditions (we remind that the stability metric $L^{-1}$ was measured in the ASL), whereby additionally stable air above augments the veer. As mentioned previously, the turning and veer near the ABL top will continue to increases for yet shallower ABLs (decreasing $h$); meanwhile $\alpha$ is less sensitive to $h$ as the upper height (used to calculate $\alpha$ and $\Delta\varphi$) exceeds the peak of the inversion-induced 'jet.' Further, such high-veer conditions are not rare for such a 'simple' site at the heights considered (60–160 m); e.g., conditions where $\Delta\varphi/\Delta z = 0.2$ and $\alpha = 0.4$ (a veer of 20° over a 100 m rotor) occur as frequently as conditions with zero shear and veer.

Towards relating veer to shear for application, we now consider the mutual behavior of $\Delta\varphi$ and $\alpha$ together at all of the sites analyzed for this work. Figure 15 shows the joint distribution of shear and veer for the sites considered, with each plot also including the conditional mean of veer per shear exponent (i.e. $\langle\Delta\varphi/\Delta z|\alpha\rangle$, as solid lines). From this figure we see a number of trends across the six cases analyzed. First, some nonlinear variation of veer with $\alpha$ is evident, along with the (less common) occurence of negative values of shear and veer, as was seen in Fig. 14 for the Høvsøre land case. Further, the veer tends to be skewed towards higher values: i.e., $\langle\Delta\varphi/\Delta z|\alpha\rangle$ exceeds the most commonly observed values of $\Delta\varphi/\Delta z$; however, the site MR does not show such skewed behavior (consistent with Fig. 10), presumably due to the complex terrain there. We note the conditional mean veer $\langle\Delta\varphi/\Delta z|\alpha\rangle$ is also more clearly nonlinear in $\alpha$, becoming less dependent on $\alpha$ in low (and negative) shear conditions; the site MR is an exception to this, with hill-induced height-dependent turning causing larger veer for $\alpha$ smaller than the most commonly-observed values there.

### 3.3.1 Simplified estimate of veer per $\alpha$

Figure 15 also includes two predictions based on the theory presented earlier. First, as discussed at the end of section 2.4, using only the shear-associated ($|S|/|G|$) portion of (14) to be practical, we arrive at the estimate

$$\frac{\partial\varphi}{\partial z} \approx \frac{|S|}{|G|}\frac{\alpha}{z} \bigg/ \sqrt{1 - \left[\frac{|S|}{|G|}\right]^2} \quad , \quad \frac{|S|}{|G|} \approx c_{s\alpha}\frac{c_{rG}}{\kappa}\frac{\ln(z/z_0)}{(\ln\mathrm{Ro}_0 - A)}; \tag{38}$$

compared to (14) the negative sign has been dropped to express the veer in coordinates commonly used in wind energy, i.e. clockwise positive. The practical form of $|S|/|G|$ in (38) employs the log-law for wind profile and reverse geostrophic drag law (32) for $u_*/|G|$; thus the constant $c_{s\alpha}$ crudely accounts for the (competing) effects of stability on both $|S|$ and the geostrophic drag (and any other mechanisms affecting $|S|/|G|$). Within the surface Rossby number $\mathrm{Ro}_0$, $G$ is calculated using (31) wherein $u_*$ is found via the log-law and $|S|$ with $z_0$; to make the plots of (38) in Fig. 15 for each site the $|S|$ is calculated per each bin of $\alpha$, with the case-specific parameters $z_0$, $f$, and height $z$ used as well. At any rate, the practical parameterization using $c_{s\alpha}$ with the log-law and (neutral) reverse GDL in (38) can roughly fit the mean conditional veer at and

**Figure 15.** Joint distribution $P(\Delta\varphi, \alpha)$ at sites considered. Solid lines: mean veer conditioned on shear exponent, $\langle\Delta\varphi/\Delta z|\alpha\rangle$; dotted lines: simple estimate via shear portion using (); dashed lines: estimate including estimate of cross-wind stress/Coriolis contribution, (). Lightest shades are 10% as likely as darkest shade in each plot.

above the most common $\alpha$ observed for the onshore sites considered ($\alpha \gtrsim 0.2$) and at $\alpha \gtrsim 0.1$ for the offshore Høvsøre case; here we have used effective roughness lengths consistent with earlier studies employing these sites ($z_0 = 1.5\,\mathrm{cm}$ for Høvsøre land, $3\,\mathrm{cm}$ for Cabauw, $0.9\,\mathrm{m}$ for Østerild, $2\,\mathrm{m}$ for MR, and $0.02\,\mathrm{cm}$ for offshore). A value of $c_{s\alpha} = 0.5$ can be seen to fit the heterogeneous terrain cases where terrain and roughness dominated over stability (Østerild and MR, bottom plots of Fig. 15), while for the more stability-dominated homogeneous cases (top plots in Fig. 15) a value of $c_{s\alpha} = 0.7$ for Høvsøre and 0.8 for Cabauw gave reasonable fits. The latter aspect could be practically addressed by directly casting $c_{s\alpha}$ as a minimal value plus an amount depending on the long-term variability in positive stability (labelled $\sigma_+$ following Kelly and Gryning, 2010); we





note Cabauw has larger values of $\sigma_+$ than Høvsøre, which has larger $\sigma_+$ then Østerild. However, obtaining such an expression is beyond the scope of the current article, and some sites could have factors other than stability which enhance the veer. We do

find that including stability within the drag law via M-O theory (for positive $L^{-1}$ values consistent with observed distributions) reduces the reverse drag-law constant by roughly 10–40% for the Rossby numbers applicable at these sites, consistent with the values of $c_{s\alpha}$ used in the plots of Fig. 15; but again, to model stability effects beyond the surface layer becomes rather complicated and is the subject of ongoing work. For reference, a value of $c_{s\alpha} = 0.6$ fits the mean veer for the Høvsøre land case during neutral condtions (not shown), in contrast to the value of 0.7 which fits when all stabilities are considered there.

### 3.3.2   Veer estimate including both $\alpha$ and cross-wind stress

We remind that for simplicity, (38) ignored the effect of cross-wind stress; it neglects not only $\langle vw \rangle$ but consequently also $\mathrm{Ro}_h$, though it does incorporate the effect of $\mathrm{Ro}_0$ seen in the simulations of Section 3.1. Thus we also consider an approximation of the $\langle vw \rangle$ terms using (33) in (14), which introduces $\mathrm{Ro}_h$, along with the parameterization for $|S|/|G|$ from (38):

$$\frac{\partial \varphi}{\partial z} \approx \frac{\frac{|S|}{|G|}\frac{\alpha}{z} + c_{vw}\frac{c_G^2}{h}\mathrm{Ro}_h}{\sqrt{1 - \left[\frac{|S|}{|G|} + c_{vw}c_G^2\mathrm{Ro}_h\right]^2}}, \qquad , \qquad \frac{|S|}{|G|} \approx c_{s\alpha}\frac{c_{rG}}{\kappa}\frac{\ln(z/z_0)}{(\ln\mathrm{Ro}_0 - A)} \tag{39}$$

where $c_G$ is found using (32), $|S|/|G|$ is calculated the same way as done earlier for (38), and $|G|$ within $\mathrm{Ro}_h$ is calculated as it was within $\mathrm{Ro}_0$ of (38). To use (39) the ABL depth must be prescribed, along with the constant $c_{vw}$ and the parameters $\{z, |S|, z_0, f\}$ also employed for (38). Given the negative curvature of lateral stress, $\partial^2\langle vw \rangle/\partial z^2 < 0$ (e.g. Wyngaard, 2010), $c_{vw}$ is negative and of order 1, with the $\langle vw \rangle$ ($\mathrm{Ro}_h$) contribution reducing the predicted veer compared to (38). With its moderating effect on the $\alpha$ contribution, the $\langle vw \rangle$ part can produce an $\alpha$-dependent 'upturn,' though slight; this is seen for the

offshore and MR cases in Fig. 15. However, the constant $c_{s\alpha}$ within $|S|/|G|$ needs to be increased in order for (39) to fit the observed $\langle \Delta\varphi/\Delta z | \alpha \rangle$; the values of 0.5 are replaced by 0.7, and $c_{s\alpha} = 0.7$ and 0.8 for Høvsøre and Cabauw are replaced by 0.8 and 0.9, respectively. The value of $c_{vw}$ giving the estimates shown in Fig. 15 was $-0.7$ for all sites, while characteristic ABL depths $h$ were taken to be $800\,\mathrm{m}$ over the simple land cases, $600\,\mathrm{m}$ offshore, and $1000\,\mathrm{m}$ over the hilly/forested terrain cases; we note that the results have limited sensitivity to $h$, but choose these values to be consistent with mean ABL depth

observations over sites of similar character and $h$ distributions aggregated by Liu and Liang (2010). One can see from Fig. 15 that the estimates of $\langle \partial\varphi/\partial z | \alpha \rangle$ using (39) are not better than the simpler form (38), though the constants $c_{s\alpha}$ and $c_{vw}$ could easily be 'tuned' together to give a better fit for each case. However, in practice one might not be able to do so, and wishes to simply predict veer based on $\alpha$; to this end, for practical applicability we suggest using (38). Though such a recommendation would appear to be neglecting $\mathrm{Ro}_h$ and the ABL depth, we note that for estimation of *mean* veer (per shear) one is not so

concerned with variations of $\mathrm{Ro}_h$ or $\mathrm{Ro}_0$ at a given site. The spread (scatter) around the mean veer seen in Fig. 15 is due to variation of stability as well as $\mathrm{Ro}_h$ or $\mathrm{Ro}_0$, and variation from site to site is also due to different distributions of $\mathrm{Ro}_h$ or $\mathrm{Ro}_0$; this is consistent with Fig. 6 and discussions following it.

   To illustrate the differences just mentioned, both the mean and standard deviation (spread) of conditional veer is shown in Fig. 16, for all the sites/cases considered . One immediately sees the character of $\langle \Delta\varphi/\Delta z | \alpha \rangle$ tends to follow the type





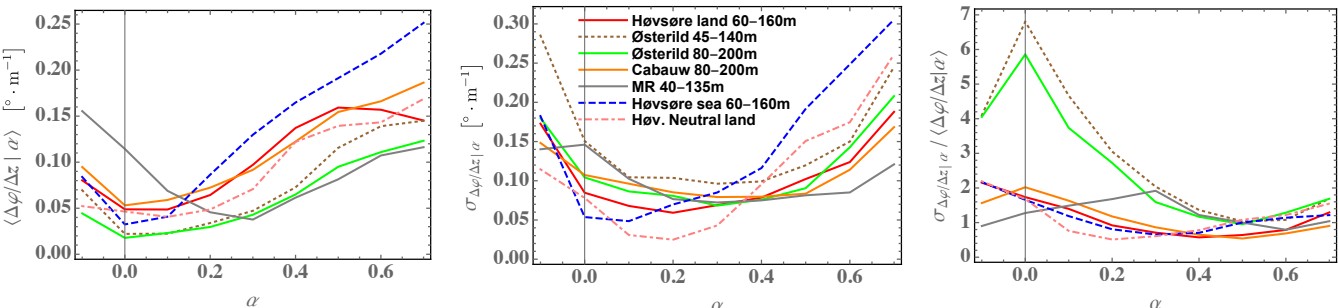

**Figure 16.** Statistics of veer conditioned on shear exponent, across sites considered.

of site; offshore has larger veer for high $\alpha$, simpler sites like Cabauw and Høvsøre onshore exhibit modest veer for large $\alpha$, and the more complex sites have more limited veer for $\alpha$ around or above its most common values of $\alpha$. But we remind that Fig. 15 shows that high-shear conditions offshore are relatively rare, and that $\alpha$ exceeding $\sim\!0.3$ is more common at the complex sites. We also see from Fig. 16 that for low-shear conditions ($\alpha < \sim 0.1$), the simpler sites exhibit higher mean veer than offshore and yet more compared to the forested cases, while much larger veer is present due to upwind hills at the MR site

for such low shear conditions (though somewhat uncommon, as seen in Fig. 15). From the middle plot we further note a that the long-term variability in veer $\sigma_{\Delta\varphi/\Delta z|\alpha}$ is lower offshore for the most commonly occuring $\alpha$ there, while veer variability does not differ so much for the most common conditions across the other sites/cases—except for the $45$–$140\,\mathrm{m}$ (lower) height range at Østerild, which shows larger veer variability due to being in the roughness sublayer above the forest there. In very high-shear conditions ($\alpha > \sim 0.5$) the veer variability is highest offshore (though rarer). However, as shown in the right-hand

plot of Fig. 16, the *relative* veer variability $\sigma_{\Delta\varphi/\Delta z|\alpha}/\langle\Delta\varphi/\Delta z|\alpha\rangle$ tends to more clearly show the different character of the sites: the spread of veer relative to its mean (conditioned on $\alpha$) is much larger in low-shear conditions over forest, while this relative spread is similar across all non-simple (forested, complex) cases for the most commonly occuring shear; the more homogeneous sites/cases exhibit comparable $\sigma_{\Delta\varphi/\Delta z|\alpha}/\langle\Delta\varphi/\Delta z|\alpha\rangle$ under most conditions. For low-shear conditions, over more complex terrain the relative veer variability decreases, departing from the inhomogeneous forested (Østerild) values due

to the large hill-induced mean veer.

The use of $\langle S|\alpha\rangle$ in the calculations was also investigated; the plots in Figs. 15–16 actually incorporated mean speed conditioned on $\alpha$, though use of each site's corresponding overall mean speed $\langle|S|\rangle$ gave nearly identical results as those shown in the plots (within 2%, not shown).

## 4   Summary and Conclusions

We have derived relationships between shear exponent ($\alpha$) and veer ($\Delta\varphi/\Delta z$), in a manner which avoids atmospheric surface-layer (ASL) assumptions about meteorological parameters; this has been done in order to be applicable at wind turbine rotor heights, regardless of whether they are within or above the ASL. Canonical behavior of veer and shear with regards to surface



roughness $z_0$ and ABL depth $h$ is also elucidated (through Rossby numbers $\text{Ro}_0$ and $\text{Ro}_h$ defined by each), through numerical solution of the 1-D RANS equations under neutral conditions with lengthscale-limited $k$-$\varepsilon$ turbulence closure (i.e. neutral but

also translatable to stable conditions, see van der Laan et al., 2020).

The derived equations and RANS results essentially show that veer most simply arises from two contributions: the shear, and the vertical variation of crosswind shear stress at a given height (mostly through $\partial^2 \langle vw \rangle / \partial z^2$, but also via $\partial \langle vw \rangle / \partial z$). The numerical RANS solutions show that the shear and crosswind-stress contributions mostly offset each other in neutral conditions, and that each is much larger (up to an order of magnitude) than the veer itself. It is further seen that $\alpha$ primarily

depends upon surface roughness in neutral conditions, with a weaker dependence on $\Delta z/h$; in contrast, $\Delta\varphi/\Delta z$ more strongly depends on the ABL depth $h$, increasing as $\text{Ro}_h^n$ where $n$ is between 1 and 1.4 for the $h$ most commonly encountered in nature (though $\Delta\varphi/\Delta z$ does also vary with $1/\ln \text{Ro}_0$). These behaviors are consistent with the shear-veer relations derived in Sec. 2.3. We note that in this work we have also derived the cause of misalignment between shear and stress, as well as its contribution to veer; we remind that RANS solutions using mixing-length type closures (as well as e.g. WRF PBL schemes which lack

turbulent transport) give stress aligned with shear, while the analytic shear-veer relations derived here allow for misalignment through the cross-wind stress.

The actual 'real-world' behavior of shear exponent and veer has also been investigated from multi-year measurements at four sites convering six different flow conditions (one with separate land and offshore sectors, one with measurements both in and above the roughness sublayer over a forest), for height spans or effective rotor diameters ranging from 47–60 m centered

around (hub) heights of 88–140 m. The observed $\{\alpha, \Delta\varphi/\Delta z\}$ include effects not fully accounted for in the equations derived here, particularly horizontal turbulent transport due to terrain inhomogeneities (Kelly, 2020) and nonstationary/transient flow conditions; though buoyancy is not *explicitly* accounted for, it primarily affects $\alpha$ and the stress, which are already incorporated into the derived veer equations.

The effect of surface-based atmospheric stability on shear and veer was examined for the relatively ideal (homogeneous)

onshore site Høvsøre, where it is seen that unstable conditions dominate the low (negative) tails of the distributions $P(\alpha)$ and $P(\Delta\varphi/\Delta z)$, while stable conditions are responsible for large $\alpha$ and $\Delta\varphi/\Delta z$; neutral conditions contributed mostly to the peaks of the shear and veer distributions. Stability efffects are consequently seen to increase the long-term variability in veer and shear, as well as veer for a given $\alpha$ — particularly for the commonly-occuring wind speeds which tend to occur below the rated speed of modern wind turbines (e.g. Kelly and Jørgensen, 2017, Appendix B). The mean of both $\alpha$ and $\Delta\varphi/\Delta z$

was larger compared to neutral conditions, due to stably stratified conditions enhancing $\alpha$ and $\Delta\varphi/\Delta z$ more than unstable conditions (we note that sites having a distribution of $1/L$ more dominated by unstable conditions, possibly some offshore, could have mean behavior similar to that found in neutral conditions).

Comparison between offshore and homogeneous onshore sectors at Høvsøre showed $\alpha$ to be smaller offshore (as one would expect), with more extreme values at higher $z$ (160 m) above the surface layer regardless of the surface; the latter is presumably

due to the effect of the capping inversion for ABL depths which occasionally approach such heights (Liu and Liang, 2010; Kelly et al., 2014b). The veer distributions also show larger values over land compared to offshore, though to a lesser extent than $P(\alpha)$; but in contrast to $\alpha$, which can increase or decrease (with wider extremes) due to the position of the jet associated



with the capping inversion, $\Delta\varphi/\Delta z$ increases overall with $z$ through the jet as the surface-based stress decreases with height
(though there can be occasional deviations from this behavior due to stress profiles affected by upwind inhomogeneities or
large coherent structures).

Two practical veer-shear relationships were derived, including parameterizations for typically unmeasured quantities con-
tained within them, then compared to the joint distributions $P(\alpha, \Delta\varphi/\Delta z)$ and the $\langle\Delta\varphi/\Delta z|\alpha\rangle$ measured from all sites over all
conditions. A simplified form (38) neglecting the stress contributions was tested, as well as one (39) containing the cross-wind
stress. Due to the relative simplicity of the practical shear-veer forms (and additional phenomena not included in them), they
needed to be calibrated in order to match observed $\langle\Delta\varphi/\Delta z|\alpha\rangle$; basically one coefficient in (38) and two in (39), all of which
were of order 1 and universal (constant) across all six sites/flow situations analyzed. The form (39) for veer including cross-
wind stress did not give a better match to observations of $\langle\Delta\varphi/\Delta z|\alpha\rangle$ across sites, compared to the simpler formula (38), and
so we recommend the latter for shear-based predictions of veer at this time. Both forms provide their best predictions (within
10% of observed) $\langle\Delta\varphi/\Delta z|\alpha\rangle$ during the most commonly-observed (moderate speeds and shear) and highest-impact (large-
veer stable) situations, with underpredictions of mean veer occuring in low-veer conditions. The observed $\langle\Delta\varphi/\Delta z|\alpha\rangle$ are
nonlinear in $\alpha$, whereas the derived forms were nearly linear, with the inclusion of cross-wind stress containing only a slight
implicit nonlinearity. Lacking turbulent transport, our predictive mean veer relations are more suited for neutral and stable
conditions where transport is less significant (e.g. Wyngaard, 2010); the underpreictions for smaller $\alpha$, dominated by unstable
conditions, evoke such. Consistent with this, the hilly site 'MR' shows yet more low-shear deviation from our predictions due
to inhomogeneity-related horizontal transport (recalling low shear means less shear-production of TKE).

Beyond the comparison of derived analytical forms with measurements of conditional mean veer $\langle\Delta\varphi/\Delta z|\alpha\rangle$, some general
trends were also noted. For a given $\alpha \gtrsim 0.2$, $\langle\Delta\varphi/\Delta z|\alpha\rangle$ was larger offshore than for the onshore cases (though we remind that
larger $\alpha$ are relatively rarer offshore compared to onshore conditions); this larger mean veer for a given $\alpha$ is due to the ABL
depth $h$ generally being lower offshore (see e.g. Liu and Liang, 2010, for offshore and onshore $h$). Perhaps counterintuitively,
over the forested site the mean veer $\langle\Delta\varphi/\Delta z|\alpha\rangle$ was smaller than other sites. As for the mean veer, for $\alpha \gtrsim 0.3$ the long-term
variability $\sigma_{\Delta\varphi/\Delta z|\alpha}$ was also found to be larger offshore; this may have impact on yaw error statistics, and may be the subject
of future research. Analogous to $\sigma_{\alpha|U}$ found in Kelly et al. (2014a) for shear, an empirical expression for the standard deviation
of veer conditioned on wind speed ($\sigma_{\Delta\varphi/\Delta z|U}$) was also found, with an approximately exponential decrease with speed.

**Ongoing and future work**

While the current work provided both theoretical meteorological relations and practical forms for veer in terms of shear, it did
so without explicit treatment of buoyancy nor turbulent transport. Some relations including stabiilty within $|S|/|G|$ in the shear
contribution to veer were developed and tested; however these were not included here, as they did not offer improvement, are
seen to be beyond the scope of the current work, and might also require stability effects to be explicitly incorporated within
the cross-stress terms. Ongoing work involves addressing the latter: i.e., self-consistent $\alpha$-based description of stability within
the veer formulations, within both the shear and cross-stress contribtions in concert with the stability-perturbed geostrophic





drag law (Arya and Wyngaard, 1975; Kelly and Troen, 2016). Future work includes incorporation terrain-induced turbulent transport parameterization (following e.g. Kelly, 2020) into the veer, as well as study of the latter via LES.

Because the veer at commonly-occuring speeds (which occur below typical rated power) and also the mean veer are larger than for commonly-assumed neutral conditions, and since we have found relations for veer variability in terms of wind speed, 725 practical ongoing work also involves vertical extrapolation of veer and accounting for its effect on power production. Accompanying this is validation and uncertainty quantification, towards pre-construction resource assessment as well as loads calculations.

*Author contributions.* MK came up with the concept, made the mathematical derivations and analysis, and wrote the text. PvdL made the RANS simulations, checked derivations, and contributed to proofreading and editing the text.

*Competing interests.* The authors declare that no competing interests are present.

*Acknowledgements.* The authors would like to acknowledge DTU Wind Energy for its internal partial support of finalizing and publishing this (old) work, through its Poul la Cour fellowship. The initial core of this work was also supported by the earlier EUDP "Demonstration of a Basis for Tall Wind Turbine Design" project, number 64011-0352. Discussions about the derivations with Mads Baungaard are sincerely appreciated.



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
