# Peer review of "From shear to veer: theory, statistics, and practical application"

_Wind Energy Science, 2022_

## Referee Comment (RC1)

The paper uses the one-dimensional, averaged governing equations of atmospheric boundary-layer flows under the assumption of horizontal homogeneity to analyse wind veer. The theory is used to derive mathematical expressions relating wind veer to wind shear and turbulent stress, and it is also used to analyse the physical mechanisms giving rise to wind veer. The theoretical results are then evaluated based on canonical RANS simulations and real observations (i.e., including for example effects of stability and heterogeneity) from 4 different sites. The authors also propose a practical approach to estimate the wind veer based on typical measurement data. The paper presents a detailed analysis of wind veer in both idealised and realistic conditions, and it provides a lot of insights and explanations for why a certain behaviour is observed. As wind veer is likely to become more and more important for wind energy purposes, I believe this work provides valuable insight into the theory and the statistics of wind veer. My main concern is that, as the paper is quite lengthy and heavy on math, more structure is needed to guide the reader. Please find below more detailed comments and suggestions as to how to further improve the manuscript.

**Main comments**

1. It is not clear from the introduction what the objectives of the study are and how it is organized. After summarizing the literature on the importance of veer, the authors quite abruptly dive into the mathematical derivations and formulations in section 2, and it took me about 20 pages until halfway section 3 to realize what the paper was actually about. The paper is quite lengthy, so more indications of how the paper is structured are needed in the introduction to guide the reader. For instance, it is by no means clear from the introduction that the authors will test the theoretical formulations of veer against RANS and observational data at 4 different sites.

2. Section 2 is quite heavy on math with a lot of different formulations for the same quantity, and it is not always clear what you are going to do with all these formulations. For example, there about 4 different formulations for the wind direction $\varphi$ (or $\cos\gamma$) (Eq. 6, 13, 25, 29) and no less than 10 for the veer $\partial\varphi/\partial z$ (or $\partial\cos\gamma/\partial z$) (Eq. 9, 10, 12, 14, 15, 16, 21, 24, 38, 39)! More structure and guidance is needed in section 2 to make sense of all the different formulations. It could for example help to indicate upfront what expression you will derive in the various subsections and for what purpose (or alternatively, e.g. for 2.3.1 and 2.3.2, what aspects you will analyse and why), and it could also be useful to provide a summary or a table with the final expression which you will (mainly) use to analyse the data in section 3.

3. Eq. 9 and 10 show that, in the absence of baroclinicity, the veer only depends on the curvature of the stress profile. Later, it is shown that veer also depends on the shear (see Eq. 12, and 14-16). How is this possible? Is it perhaps so that the shear affects the veer by its impact on the stress

profile, so that Eq.9-10 are implicitly dependent on the shear? Please comment on the physics explaining these different forms in the paper.

4. The authors propose two practical equations to find veer based on shear, equation 38 and 39. Eq. 38 only accounts for the shear, while Eq. 39 includes the contribution from the cross-wind stress. In Fig. 15 it is shown that both equations give comparable results, and for simplicity equation 38 is recommended. However, this seems to contradict with the earlier finding from fig 5, showing that the contributions of shear and crosswind stress are an order of magnitude larger than the veer and mostly balance each other. How is it then possible to get good results with Eq. 18 in which the crosswind stress is neglected? Please comment on thing apparent contradiction in the paper.

**Other scientific comments**

1. Line 98: by definition $B = -z/L$. I believe this is incorrect and should be $B/\varepsilon_0 = -z/L$.

2. Eq. 8: It seems to me that you only use the first form of $d\varphi$ for later derivation and interpretation. The derivation of the second and third form is not so straightforward, and, given that these forms are not being used (as far as I can tell), I think they make the math in this section unnecessary complicated. Moreover, I believe the second form (with $S$ and $S^*$) is missing a minus sign (or equivalently, the factor $i$ should be in the denominator instead of in the numerator). It is not clear to me how you derive the third form from the second one. Unless these complex forms are essential for later derivation or interpretation, consider removing them.

3. The wording used to introduce and derive equations in section 2.3 is not always clear. For example, line 143, what do you mean with "taking alternately the time derivative of (8)"? Line 152: what do you mean with the "dimensionless deviation of the wind from streamwise"? Line 168: "We note that (14) and (15) are more direct alternatives to dealing with functions of $\varphi_G$". Not clear to me what this is about. It is also not clear how Eq. 16 is obtained.

4. I'm confused about Eq. 17: The text says that $\beta_{\mathrm{ma}}$ is the angle between $\partial S/\partial z$ and $\langle sw \rangle$, but Eq. 17 uses $\arg(S)$. Are you implying that $\arg(S) = \arg(\partial S/\partial z)$? I don't see how this holds mathematically. Please explain.

5. Line 192: Please add a reference for the Rotta parametrization.

6. Eq. 18 is introduced to show the root of the misalignment between shear and stress, but it is not entirely clear to me from the surrounding text how this equation explains the root of the stress-shear misalignment. Line 197-199 only talks about the absence of misalignment when the flux-gradient relation is used, but a clear interpretation of Eq. 18 and how it explains

misalignment is missing. Are you saying that misalignment is directly related to the turbulent transport of turbulent stress (the last term in Eq. 18)?

7. Section 2.3.2: I found the section title misleading. The section does not talk about alignment of shear and stress, but instead discusses two canonical solutions for the veer and the relation with shear.

8. Eq. 21: How is this equation obtained? Do you get it from Eq. 10? Moreover, on line 222 you mention it "defies analytical solution", but I'm not sure what you mean with that. The differential equation has two unknowns $\varphi(z)$ and $U(z)$ so you need an additional equation anyway to solve it.

9. Line 243: It would be more clear to indicate in the subsection title that this is the Ellison regime you referred to earlier. The current subsection title "Linear diffusivity profile: ..." is not clear and I had to search for where this Ellison regime is actually discussed.

10. Fig. 1 left plot is not entirely clear to me. Which part of the curve is for low/high values of $z$? Line 260 says that you can see that "the Ekman solution produces less mixing away from the surface ... and consequently a higher shear exponent than Ellison's." Is this statement referring to the left plot? Difficult to appreciate without knowing where low/high values of $z$ are in the plot.

11. Where is Eq. 35 coming from? Please provide a reference.

12. Line 375: How did you find the exponent -1.4? Can this be derived mathematically from Eq. 33 or is this an empirical estimate based on visual observation?

13. Line 429-430: "... follow a curve which resembles the nondimensional M-O shear function ...". Not all readers will know the shape of the $\Phi_M$ curve by hard. Please show the curve or provide a reference.

14. Line 450: Not clear what the angular brackets in $\langle \alpha \rangle$ mean.

15. Line 487: "The lowest [veer scale] corresponds to the offshore Høvsøre case, while the highest ...". It is not particularly clear how this can be seen. The veer scale defines the slope of the PDF in log scale, correct? As in, the higher the veer scale, the lower the slope?

**Minor/technical comments**

1. Variable $Ro_0$ is first used on line 264 but only defined on line 290. Please define upon first use.

2. Line 274: Should $|S/G|$ be $|S|/|G|$?

3. Line 307: "along with  (30) or (32)..."

4. Line 374: "... that is not inconsistent with" $\Rightarrow$ Avoid double negation and just use "consistent with"

5. Probability density function values are sometimes difficult to interpret. For example, in figure 7 (right), a value of $P(\alpha, L^{-1}) = 1000$ is not very meaningful unless you know the bin size $d\alpha$ and $dL^{-1}$. It would be much clearer if you could show results in terms of their relative frequency of occurrence.

6. Caption of Figure 15: what are the two empty brackets "()"? Should these contain a reference to an equation?

7. Line 604: "condtions" $\Rightarrow$ "conditions"

8. Line 667: "convering" $\Rightarrow$ "covering"?

9. Line 703: "underpreictions" $\Rightarrow$ "underpredictions"

10. Line 716: "stabiilty" $\Rightarrow$ "stability"

---

## Referee Comment (RC2)

**Review of "From Shear to Veer: Theory, Statistics and Practical Applications"**

April 2023

**1 Overview**

This paper is a very large body of work trying to connect the veer to shear for wind energy applications. This is very relevant for modern and next generation of turbines that will be very large and operating outside the surface layer of the atmospheric boundary layer (ABL). The first part of this paper is highly theoretical followed by comparisons to RANS simulations of the ABL. The latter part of this paper analyzes the experimental data from several wind farms and tries to connect the shear and veer based on the derivations in the theoretical sections. This is very good work and deserves to be published. However, I feel that there's likely a major flaw in Equation 14 that is the basis for Equations 38 and 39 that connect to the theme and the title of the paper. This is outlined in further detail in the next section under "Page 7, Line 165". If the authors could address this and the other specific comments in the next section, this paper should be published and would prove a great addition to the literature.

**2 Specific comments**

**Page 4, Line 88** What is the purpose of Section 2.1.1? I'm not sure where this discussion is used in the rest of the paper. Also, since this paper considers non-zero veer, the balance of terms in the turbulent kinetic energy equation will be

$$\frac{\mathrm{d}e}{\mathrm{d}t} = 0 = -\langle uw \rangle \frac{\mathrm{d}U}{\mathrm{d}z} - \langle vw \rangle \frac{\mathrm{d}V}{\mathrm{d}z} + B + T - \epsilon.$$

Since $\mathrm{d}U/\mathrm{d}z \neq \mathrm{d}|S|/\mathrm{d}z$, I think the expression in Equation 4 of the manuscript will no longer hold.

**Page 5, Line 132** I'm not sure of the purpose of the expressions with the complex math as they're not used further in the manuscript. However,

$$d\varphi = \frac{U \, \mathrm{d}V - V \, \mathrm{d}U}{|S|^2} = -i \frac{S^* \mathrm{d}S - S \mathrm{d}S^*}{2|S|^2},$$

where the manuscript uses $i$ instead of $-i$ in the second term.

**Page 7, Line 165** The derivative of Equation 13 is taken here after the evaluation at the height of interest to arrive at Equation 14. I don't think this is correct. If you followed the math described at the top of Page 7, it will instead be

$$\gamma = \cos^{-1}\left[\frac{|S|}{|G|} + \frac{1}{f|S||G|}\left(U\frac{\partial\langle vw\rangle}{\partial z} - V\frac{\partial\langle uw\rangle}{\partial z}\right)\right],$$

$$\frac{\partial\gamma}{\partial z} = \left(-\frac{|S|\alpha}{|G|z} - \left(U\frac{\partial\langle vw\rangle}{\partial z} - V\frac{\partial\langle uw\rangle}{\partial z}\right)\frac{1}{f|S||G|}\frac{\alpha}{z}\right.$$
$$\left. -\frac{1}{f|S||G|}\left(\frac{\partial U}{\partial z}\frac{\partial\langle vw\rangle}{\partial z} - \frac{\partial V}{\partial z}\frac{\partial\langle uw\rangle}{\partial z} + U\frac{\partial^2\langle vw\rangle}{\partial z^2} - V\frac{\partial^2\langle uw\rangle}{\partial z^2}\right)\right)$$
$$\left[1 - \left(\frac{1}{f|G|}\frac{\partial\langle vw\rangle}{\partial z}\perp + \frac{|S|}{|G|}\right)^2\right]^{-1/2},$$

$$= \left(-\frac{|S|\alpha}{|G|z} - \frac{\alpha}{f|G|z}\frac{\partial\langle vw\rangle}{\partial z}\perp - \frac{1}{f|S||G|}\left(\frac{\partial U}{\partial z}\frac{\partial\langle vw\rangle}{\partial z} - \frac{\partial V}{\partial z}\frac{\partial\langle uw\rangle}{\partial z} + U\frac{\partial^2\langle vw\rangle}{\partial z^2}\right)\right)$$
$$\left[1 - \left(\frac{1}{f|G|}\frac{\partial\langle vw\rangle}{\partial z}\perp + \frac{|S|}{|G|}\right)^2\right]^{-1/2}.$$

Out of the 5 terms in the numerator of the last expression, you only have terms 1 and 5 in Equation 15. I'm not sure how the terms 2, 3 and 4 are eliminated. Could you please explain/describe the math here.

Also, what is the connection to Equation 10 of the manuscript here. That appears to be far simpler, albeit lacking any shear exponent!

**Page 10, Line 255** Typo in "moreso"? Should be "more so"?

**Page 12, Line 294** The velocity profiles based on the shear exponent and the log-law are

$$|S| = |S|_{\text{ref}}\left(\frac{z}{z_{\text{ref}}}\right)^\alpha, \text{and}$$
$$|S| = \frac{u_*}{\kappa}\ln\left(\frac{z}{z_0}\right)$$

respectively. There is quite a bit of discussion on Page 3 Lines 80-87 describing the invalidity of the log-law above the surface layer and how the shear exponent is better suited for wind energy applications. How do you explain using the log law expression again, especially without the corrections for stability as in M-O law?

**Page 12, Lines 296-302** The estimate for the first vertical derivative of the Reynolds stresses seems reasonable as $\partial\langle uw\rangle/\partial z \approx u_*^2/h$, although the sign is likely negative. However, the estimate for the second derivative is not very clear. In addition, considering the mean momentum equation from Equation 7,

$$\frac{\partial\langle vw\rangle}{\partial z} = -f(U - U_G),$$
$$\frac{\partial^2\langle vw\rangle}{\partial z^2} = -f\frac{\partial U}{\partial z} = -f\alpha\frac{U}{z}.$$

This tells me the sign of this constant $c_{vw}$ must be negative. This might make the numerator of Equation 14 almost zero, suggesting other terms are dominating as expressed in my concerns for Page 7, Line 165. Could you please expand the explanation for your estimate of the second derivative of the stresses?

**Page 12, Line 306** Based on the expressions in Equation 31, I got this to be

$$\sin\varphi_G = \frac{-c_G}{\kappa}\left\{B\cos\varphi_0 - \left[\ln\frac{u_*}{fz_0} - A\right]\sin\varphi_0\right\}. \tag{1}$$

The multiplication of $B$ by $\cos\varphi_0$ is missing. Could you please check again.

**Page 12, Line 315** A summary of the expressions to be evaluated after substitutions listed in Section 2.4 would be extremely helpful before evaluating them in Section 3.1.

**Page 14, Line 355** A more thorough overview of how the RANS simulations covers the space of interest would be helpful here.

**Page 14, Line 358** How does the expression in Equation 14 compare to Equation 9 in the right plot in Figure 2? Based on my concerns expressed for Page 7, Line 165, I expect the correlation to be not as good as for Equation 9.

**Page 15, Line 374** Behavior that is "not inconsistent"? Do you mean consistent? There is no expression derived in Section 2 that gives $Ro_h^1.4$. This is also likely affected by concerns for Page 7, Line 165.

**Page 26, Line 583 and Page 28, Line 609** This simplification in Equations 38 and 39 are likely in error. Could you please check my concerns for Page 7, Line 165 before confirming these results.

**Page 29, Figure 16** The standard deviation of the veer is almost the same or much larger than the mean from the experiments. It would be hard to consider this as proof of the expressions derived in Equations 38 and 39.

---

## Author Comment (AC1)

**Response to Referee Comment 'RC1'**

Mark Kelly     M. Paul van der Laan

March 20, 2023

This is the authors' response to Dries Allaerts' (RC1) review of our article **"From shear to veer: theory, statistics, and practical application"** (WES 2022-119). We include the referee's comments *in italic script*, followed point-wise by our responses in blue.

The authors are thankful for the time and effort spent—and constructive criticism made—by the reviewer.

**Overall review**

*The paper uses the one-dimensional, averaged governing equations of atmospheric boundary-layer flows under the assumption of horizontal homogeneity to analyse wind veer. The theory is used to derive mathematical expressions relating wind veer to wind shear and turbulent stress, and it is also used to analyse the physical mechanisms giving rise to wind veer. The theoretical results are then evaluated based on canonical RANS simulations and real observations (i.e., including for example effects of stability and heterogeneity) from 4 different sites. The authors also propose a practical approach to estimate the wind veer based on typical measurement data. The paper presents a detailed analysis of wind veer in both idealised and realistic conditions, and it provides a lot of insights and explanations for why a certain behaviour is observed. As wind veer is likely to become more and more important for wind energy purposes, I believe this work provides valuable insight into the theory and the statistics of wind veer. My main concern is that, as the paper is quite lengthy and heavy on math, more structure is needed to guide the reader. Please find below more detailed comments and suggestions as to how to further improve the manuscript.*

The authors appreciate the constructive comments and overall suggestion here; although we removed quite a bit of material (and math) for submission, you have helped to show how it can be more readable, understandable, and ultimately usable through more explicit indication of its structure and intent.

**Main comments**

1. *It is not clear from the introduction what the objectives of the study are and how it is organized. After summarizing the literature on the importance of veer, the authors quite abruptly dive into the mathematical derivations and formulations in section 2, and it took me about 20 pages until halfway section 3 to realize what the paper was actually about. The paper is quite lengthy, so more indications of how the paper is structured are needed in the introduction to guide the reader. For instance, it is by no means clear from the introduction that the authors will test the theoretical formulations of veer against RANS and observational data at 4 different sites.*

   We have adjusted the introduction to clarify the structure of the paper and its intent. The last paragraph of section 1 has been expanded to include this:

   *"In this paper we investigate wind veer, showing its joint behaviors with and connection to shear and key parameters used to describe atmospheric boundary layer flow. In Sec. 2, after reviewing expression of the shear exponent and its relation to stability and turbulence, we derive new relations for veer; we show veer to be composed of shear-driven and Coriolis-associated stress gradient contributions. The theoretical behavior of veer is also derived for canonical cases such as Ekman and surface-layer flow, as well as the effect of shear-stress misalignment on veer. Further, in Sec. 2.4 practical relations from micrometeorology are elucidated, towards evaluation of the expressions developed for veer. Section 3*

*includes analysis of veer, exploring and connecting the developed relations to both computational modelling and observations. Section 3.1 gives RANS (mean) simulation results over flat terrain in neutral conditions for hundreds of combinations of surface-Rossby number and ABL-depth Rossby number, showing the dependence of veer on the latter as well as the counteracting behavior of veer's two primary components. Section 3.2 begins with analysis of multi-year observations from six different flow regimes across four sites showing the statistical behavior of shear with stability, and subsequently that of veer, also providing new empirical relations for the probability of occurrence of larger veer (due to the effect of stable conditions) and for the variability of veer with wind speed. The observational analysis concludes in Sec. 3.3 with simplified practical relations for veer based on observed shear, including comparison with joint distributions of veer and shear across the six flows analyzed. Finally the results summarily discussed and conclusions given, with ongoing and future work also described for the reader."*

2. *Section 2 is quite heavy on math with a lot of different formulations for the same quantity, and it is not always clear what you are going to do with all these formulations. For example, there about 4 different formulations for the wind direction $\varphi$ (or $\cos\gamma$) (Eq. 6, 13, 25, 29) and no less than 10 for the veer $\partial\varphi/\partial z$ (or $\partial\cos\gamma/\partial z$) (Eq. 9, 10, 12, 14, 15, 16, 21, 24, 38, 39)! More structure and guidance is needed in section 2 to make sense of all the different formulations. It could for example help to indicate upfront what expression you will derive in the various subsections and for what purpose (or alternatively, e.g. for 2.3.1 and 2.3.2, what aspects you will analyse and why), and it could also be useful to provide a summary or a table with the final expression which you will (mainly) use to analyse the data in section 3.*

   Along with the changes to the introduction section, following your suggestion we have added an explanatory paragraph at the beginning of section 2 to include guidance on the formulations:

   *"In this section we define the shear exponent and veer, then derive relations for veer in terms of shear and vertical gradients of stress, as mentioned in the previous paragraph. Section 2.3 provides a number of expressions for veer; this is done to facilitate its calculation and interpretation in the different coordinate systems typically considered in wind energy flow analyses, and we also include forms that are independent of coordinate system. Because coordinates aligned with the mean wind for a given height of interest (e.g. hub height) are commonly used in wind energy, and because expressions for veer in such a coordinate system are simpler to express and calculate, we ultimately arrive at two forms in such a system (eqs. 14 and 16); due to its robustness, one of these (eq. 14) will later be shown in section 3.3 to be further simplifiable and usable (as eq. 38 or 39) in comparison with measurements."*

   However, we note that (21) and (24) are not just generic veer expressions; also, (25) and (29) are not formulations for the wind direction but are angular differences $\Delta\varphi$ for the canonical Ekman and Ellison cases. We have updated section 2.3.2 (which includes eqs. 21–29) to more clearly and consistently include subscripts distinguishing particular cases (e.g. Ekman or Ellison) of the veer or wind profile, so that upon browsing the reader will not be confused or overwhelmed by what might otherwise appear as "yet another way to write veer."

3. *Eq. 9 and 10 show that, in the absence of baroclinicity, the veer only depends on the curvature of the stress profile. Later, it is shown that veer also depends on the shear (see Eq. 12, and 14–16). How is this possible? Is it perhaps so that the shear affects the veer by its impact on the stress profile, so that Eq. 9–10 are implicitly dependent on the shear? Please comment on the physics explaining these different forms in the paper.*

   Yes, these expressions implicitly include shear, as we now more explicitly point out below eq. 10; i.e., we add the text *"since the shear is implicit in the stress terms (and one would need to know the profiles of horizontal stresses to use these equations)"*.

4. *The authors propose two practical equations to find veer based on shear, equation 38 and 39. Eq. 38 only accounts for the shear, while Eq. 39 includes the contribution from the cross-wind stress. In Fig. 15 it is shown that both equations give comparable results, and for simplicity equation 38 is recommended. However, this seems to contradict with the earlier finding from fig 5, showing that the contributions*

*of shear and crosswind stress are an order of magnitude larger than the veer and mostly balance each other. How is it then possible to get good results with Eq. 18 in which the crosswind stress is neglected? Please comment on thing apparent contradiction in the paper.*

Basically the shear and crosswind stress contributions act similarly, which is part of how they nearly cancel each other out, leaving the veer as a sort of residual difference. The practical form (38) essentially just picks the 'easier' term to model. Admittedly we did not explain this in section 3, so now we add this around eq. 38 (also as motivation for such). E.g., the coefficient $c_{s\alpha}$ crudely accounts for this (in addition to that of stability, which was already written); the new/augmented text reads: *"The basis for the simple shear-driven form can be understood by recalling section 3.1 and Fig. 5, where we showed that the shear and crosswind stress-curvature contributions behaved in nearly identical but opposite fashions, with their sum amounting to $d\cos\gamma/dz$; (38) can be considered as a simple model assuming the veer behaves like either of its two components, but simply smaller in magnitude. The practical form of $|S|/|G|$ in (38) employs the log-law for wind profile and reverse geostrophic drag law (32) for $u_*/|G|$. The constant $c_{s\alpha}$ crudely accounts for the (competing) effects of stability on both $|S|$ and the geostrophic drag (and any other mechanisms affecting $|S|/|G|$), but also accounts for the smaller magnitude of $\partial\varphi/\partial z$ compared to its shear-driven component"*.

**Other scientific comments**

1. Line 98: by definition $B = -z/L$. I believe this is incorrect and should be $B/\epsilon_0 = -z/L$.

   Yes, corrected.

2. *Eq. 8: It seems to me that you only use the first form of $\varphi$ for later derivation and interpretation. The derivation of the second and third form is not so straightforward, and, given that these forms are not being used (as far as I can tell), I think they make the math in this section unnecessary complicated. Moreover, I believe the second form (with $S$ and $S^*$) is missing a minus sign (or equivalently, the factor $i$ should be in the denominator instead of in the numerator). It is not clear to me how you derive the third form from the second one. Unless these complex forms are essential for later derivation or interpretation, consider removing them.*

   Yes, we originally had a number of complementary derivations which were cast in complex form, using the second and third parts, but removed them before submission; we forgot to remove the complex pieces from (8), but have now done so.

3. *The wording used to introduce and derive equations in section 2.3 is not always clear. For example, line 143, what do you mean with "taking alternately the time derivative of (8)"? Line 152: what do you mean with the "dimensionless deviation of the wind from streamwise"? Line 168: "We note that (14) and (15) are more direct alternatives to dealing with functions of $\varphi_G$". Not clear to me what this is about. It is also not clear how Eq. 16 is obtained.*

   Regarding 'alternately', this referred to deriving veer by using a different differentiation than that which had just been shown for the preceeding expressions; we have now clarified the text. On line 152, 'dimensionless' referred to the speed; this has been corrected. Regarding line 168: we have simplified to be more direct, as $\varphi_G$ arises from the definition of $\gamma$.
   We introduce how to derive (16), but did not include the full details of its derivation because there are already so many equations in the document; we have added the word 'then' connecting the two sentences introducing (16), to better indicate the order of re-expressing terms following (12).

4. *I'm confused about Eq. 17: The text says that $\beta_{ma}$ is the angle between $\partial S/\partial z$ and $\langle sw\rangle$, but Eq. 17 uses arg(S). Are you implying that $\arg(S) = \arg(\partial S/\partial z)$? I don't see how this holds mathematically. Please explain.*

   We have fixed the typographic error in Eq. 17.

5. Line 192: Please add a reference for the Rotta parametrization.

   We chose to just use the Wyngaard (2004) reference because it more clearly shows the Rotta form as commonly used, and because Rotta (1951) is in German; however, we have now added the latter reference following your suggestion.

6. *Eq. 18 is introduced to show the root of the misalignment between shear and stress, but it is not entirely clear to me from the surrounding text how this equation explains the root of the stress-shear misalignment. Line 197–199 only talks about the absence of misalignment when the flux-gradient relation is used, but a clear interpretation of Eq. 18 and how it explains misalignment is missing. Are you saying that misalignment is directly related to the turbulent transport of turbulent stress (the last term in Eq. 18)?*

   Indeed misalignment can arise due to the last term in (18); lines 197–199 further mention that advection and horizontal transport (in addition to vertical stress transport) can cause misalignment. Further interpretation (with accompanying derivations) is the subject of ongoing work/writing beyond the scope of this article.

7. *Section 2.3.2: I found the section title misleading. The section does not talk about alignment of shear and stress, but instead discusses two canonical solutions for the veer and the relation with shear.*

   This sub-subsection does have a deprecated title, due to re-organization and reduction before submission. It is now re-titled "Canonical solutions using an eddy diffusivity".

8. *Eq. 21: How is this equation obtained? Do you get it from Eq. 10? Moreover, on line 222 you mention it "defies analytical solution", but I'm not sure what you mean with that. The differential equation has two unknowns $\varphi(z)$ and $U(z)$ so you need an additional equation anyway to solve it.*

   We have now included this information and re-worded for clarity.

9. *Line 243: It would be more clear to indicate in the subsection title that this is the Ellison regime you referred to earlier. The current subsection title "Linear diffusivity profile: …" is not clear and I had to search for where this Ellison regime is actually discussed.*

   The paragraph title has now been renamed "Ellison solution (linear diffusivity profile / surface-layer regime)".

10. *Fig. 1 left plot is not entirely clear to me. Which part of the curve is for low/high values of $z$? Line 260 says that you can see that "the Ekman solution produces less mixing away from the surface … and consequently a higher shear exponent than Ellison's." Is this statement referring to the left plot? Difficult to appreciate without knowing where low/high values of $z$ are in the plot.*

   In the left-hand plot, near-surface values are near the top, i.e. giving the highest values of $\alpha$ and $h\partial\varphi/\partial z$; we now note this in the text to clarify. The sentence of line 260 has also been cleaned up with regard to this, and to reflect what is seen from the different plots of Fig. 1.

11. *Where is Eq. 35 coming from? Please provide a reference.*

   It arises in our earlier (2020) paper; such reference is now included.

12. *Line 375: How did you find the exponent -1.4? Can this be derived mathematically from Eq. 33 or is this an empirical estimate based on visual observation?*

   We find it by 'empirically' by fitting to the output; this aspect is now included in the text.

13. *Line 429-430: "... follow a curve which resembles the nondimensional M-O shear function ...". Not all readers will know the shape of the $\Phi_M$ curve by hard. Please show the curve or provide a reference.*

We now include the $\alpha$-analogue to $\Phi_m(z/L)$ in the plot and text.

14. *Line 450: Not clear what the angular brackets in $\langle \alpha \rangle$ mean.*

This denotes the mean, as indicated later in the sentence adjectively for veer; however we now explicitly indicate this also before $\langle \alpha \rangle$.

15. *Line 487: "The lowest [veer scale] corresponds to the offshore Høvsøre case, while the highest ...". It is not particularly clear how this can be seen. The veer scale defines the slope of the PDF in log scale, correct? As in, the higher the veer scale, the lower the slope?*

Since the veer scale is defined as $\Upsilon_{\text{veer}}^{-1} \equiv \partial[\ln P(\Delta\varphi/\Delta z)]/\partial(\Delta\varphi/\Delta z)$, i.e. via reciprocal, then smaller $\Upsilon_{\text{veer}}$ correspond to steeper (more negative) log-slopes; the offshore case (blue dashes in Fig. 10) are the steepest, with $\Upsilon_{\text{veer}} = 0.07°/\text{m}$.

**Minor/technical comments**

1. *Variable $Ro_0$ is first used on line 264 but only defined on line 290. Please define upon first use.*

This has been rectified.

2. *Line 274: Should $|S/G|$ be $|S|/|G|$?*

Yes; this typo has been corrected.

3. *Line 307: "along with with (30) or (32)..."*

The extra 'with' has been removed.

4. *Line 374: "... that is not inconsistent with" → Avoid double negation and just use "consistent with"*

Done.

5. *Probability density function values are sometimes difficult to interpret. For example, in figure 7 (right), a value of $P(\alpha, L^{-1}) = 1000$ is not very meaningful unless you know the bin size $d\alpha$ and $dL^{-1}$. It would be much clearer if you could show results in terms of their relative frequency of occurrence.*

We include frequency of occurrence in the captions to the plots where this was lacking (under Figures 7, 9, 13, and 14).

6. *Caption of Figure 15: what are the two empty brackets "()"? Should these contain a reference to an equation?*

The equation references disappeared, and are now corrected/included.

7. *Line 604: "condtions" → "conditions"*

Corrected.

8. *Line 667: "convering" → "covering"?*

Corrected.

9. *Line 703: "underpreictions" → "underpredictions"*

   Corrected.

10. *Line 716: "stabiilty" → "stability"*

    Corrected.

Again we express appreciation to the Dries Allaerts for the review, which served to ultimately improve the scientific dissemination and support the scientific method at large.

with kind regards,

Mark Kelly and M. Paul van der Laan
Department of Wind and Energy Systems
Risø Lab/Campus, Danish Technical University

---

## Author Comment (AC2)

**Response to Referee Comment 'RC2'**

Mark Kelly        M. Paul van der Laan

April 14, 2023

This is the authors' response to Ganesh Vijayakumar's review (RC2) of our manuscript **"From shear to veer: theory, statistics, and practical application"** (WES 2022-119). We include the referee's comments *in italic script*, followed point-wise by our responses in blue.

The authors are thankful for the time and effort spent—and constructive criticism made—by the reviewer.

**Overall review ('*overview*')**

*This paper is a very large body of work trying to connect the veer to shear for wind energy applications. This is very relevant for modern and next generation of turbines that will be very large and operating outside the surface layer of the atmospheric boundary layer (ABL). The first part of this paper is highly theoretical followed by comparisons to RANS simulations of the ABL. The latter part of this paper analyzes the experimental data from several wind farms and tries to connect the shear and veer based on the derivations in the theoretical sections. This is very good work and deserves to be published. However, I feel that there's likely a major flaw in Equation 14 that is the basis for Equations 38 and 39 that connect to the theme and the title of the paper. This is outlined in further detail in the next section under "Page 7, Line 165". If the authors could address this and the other specific comments in the next section, this paper should be published and would prove a great addition to the literature.*

The authors again thank the reviewer for these positive comments. We disagree that there is a 'major flaw' in equation 14, but rather a small factor was not included. We note it was also confirmed by the RANS simulations, and respond in detail to this specific point below, including a minor update/correction to (14) and explanation in the text as well as corresponding update to eq.39 (which is now 40; eq.38 was not affected). We address the other specific comments of the reviewer below as well.

**Specific comments**

- ***Page 4, Line 88:*** *What is the purpose of Section 2.1.1? I'm not sure where this discussion is used in the rest of the paper.*

  As expressed in the reply to reviewer 1 (RC1), we have added to and modified the introduction section, to clarify the intent and structure of this work; this includes motivating section 2.1.1. Specifically, we now state "*In Sec. 2, after reviewing expression of the shear exponent and its relation to stability and turbulence, we derive new relations for veer;*" the relation of shear to stability connects to/informs the analysis of observed $\alpha$ with stability in Section 3.2.

  *Also, since this paper considers non-zero veer, the balance of terms in the turbulent kinetic energy equation will be*
  $$\frac{\mathrm{d}e}{\mathrm{d}t} = 0 = -\langle uw\rangle\frac{\mathrm{d}U}{\mathrm{d}z} - \langle vw\rangle\frac{\mathrm{d}V}{\mathrm{d}z} + B + T - \epsilon.$$
  *Since $\mathrm{d}U/\mathrm{d}z \neq \mathrm{d}|S|/\mathrm{d}z$, I think the expression in Equation 4 of the manuscript will no longer hold.*

  Regarding the TKE budget, the text continuing just below Equation 4 states "for a given height $z$, where the streamwise direction is defined by the mean wind $U(z)$ and we have suppressed $z$-dependences for brevity", so that $\langle vw\rangle dV/dz \to 0$ in the surface layer for such coordinates. However,

you are correct in that a cross-wind term will linger; the expression (4) for shear exponent becomes

$$\alpha = \frac{z}{U} \frac{\left(\varepsilon - B - T + \langle vw \rangle \frac{dV}{dz}\right)}{-\langle uw \rangle}.$$

This will not qualitatively affect the effect of buoyancy on shear exponent, but is now addressed in the text. In section 2.1.1 we add *"We point out that Kelly et al. (2014a) ignored cross-wind stress $\langle vw \rangle$ when deriving (4), however it still shows that shear will increase in stable conditions ($B < 0$) and decrease in unstable conditions ($B > 0$), as will be demonstrated using observations in Sec. 3.2; further, as we will see in section 2.3, this is also related to the veer."* At the end of section 2.3.0 (just before §2.3.1) we also add a paragraph which includes a new equation (eq.17) updating the form of eq.4 to include the lateral (veer-associated) contribution: *"One last relation between shear and veer can also be elucidated, by considering a corrected version of (4). By keeping the lateral shear term $\langle vw \rangle \partial V / \partial z$ in the TKE rate equation, then again using coordinates defined wth x in the mean direction at height z and subsequently $\partial V / \partial z \to U \partial \varphi / \partial z$, then (4) contains an additional contribution, becoming*

$$\alpha|_{\mathbf{e}_x \parallel \mathbf{U}(z)} = \frac{\varepsilon - B - T}{-U \langle uw \rangle_\parallel / z} - z \frac{\partial \varphi}{\partial z} \frac{\langle vw \rangle_\perp}{\langle uw \rangle_\parallel}. \tag{17}$$

*Recalling in the ABL that $\langle uw \rangle_\parallel < 0$ (momentum gets transferred towards the surface), because $\langle vw \rangle_\perp > 0$ in the ABL (Wyngaard, 2010) we see as in (14)–(16) that negative $\partial \varphi / \partial z$ (clockwise veer) is associated with positive shear; we remind that the sign of $\partial \varphi / \partial z$ is flipped in typical wind energy coordinates (left-handed, with 0 degrees corresponding to wind from the north and increasing clockwise). Although we have provided (17) to both improve (4) from Kelly et al. (2014a) and offer insight into how shear and veer are linked within the context of TKE, we advise that it is not easily utilized compared to forms like (14); the latter will be applied and investigated further in later sections."*

- **Page 5, Line 132:** *I'm not sure of the purpose of the expressions with the complex math as they're not used further in the manuscript. However,*

$$\mathrm{d}\varphi = \frac{U \, \mathrm{d}V - V \, \mathrm{d}U}{|S|^2} = -i \frac{S^* \, \mathrm{d}S - S \, \mathrm{d}S^*}{2|S|^2},$$

*where the manuscript uses $i$ instead of $-i$ in the second term.*

As also mentioned in the authors' reply (AC1) to reviewer 1, we have corrected the minus sign error and also removed the last (complex) expression on the right-hand side due to it no longer being used.

- **Page 7, Line 165:** *The derivative of Equation 13 is taken here after the evaluation at the height of interest to arrive at Equation 14. I don't think this is correct. If you followed the math described at the top of Page 7 , it will instead be*

$$\gamma = \cos^{-1}\left[ \frac{|S|}{|G|} + \frac{1}{f|S||G|}\left(U \frac{\partial \langle vw \rangle}{\partial z} - V \frac{\partial \langle uw \rangle}{\partial z}\right)\right]$$

$$\frac{\partial \gamma}{\partial z} = \left(-\frac{|S|\alpha}{|G|z} - \left(U \frac{\partial \langle vw \rangle}{\partial z} - V \frac{\partial \langle uw \rangle}{\partial z}\right)\frac{1}{f|S||G|}\frac{\alpha}{z}\right.$$
$$\left. - \frac{1}{f|S||G|}\left(\frac{\partial U}{\partial z}\frac{\partial \langle vw \rangle}{\partial z} - \frac{\partial V}{\partial z}\frac{\partial \langle uw \rangle}{\partial z} + U \frac{\partial^2 \langle vw \rangle}{\partial z^2} - V \frac{\partial^2 \langle uw \rangle}{\partial z^2}\right)\right)$$
$$\left[1 - \left(\frac{1}{f|G|}\frac{\partial \langle vw \rangle_\perp}{\partial z} + \frac{|S|}{|G|}\right)^2\right]^{-1/2}$$
$$= \left(-\frac{|S|\alpha}{|G|z} - \frac{\alpha}{f|G|z}\frac{\partial \langle vw \rangle_\perp}{\partial z} - \frac{1}{f|S||G|}\left(\frac{\partial U}{\partial z}\frac{\partial \langle vw \rangle}{\partial z} - \frac{\partial V}{\partial z}\frac{\partial \langle uw \rangle}{\partial z} + U \frac{\partial^2 \langle vw \rangle}{\partial z^2}\right)\right)$$
$$\left[1 - \left(\frac{1}{f|G|}\frac{\partial \langle vw \rangle_\perp}{\partial z} + \frac{|S|}{|G|}\right)^2\right]^{-1/2}.$$

*Out of the 5 terms in the numerator of the last expression, you only have terms 1 and 5 in Equation 15. I'm not sure how the terms 2, 3 and 4 are eliminated. Could you please explain/describe the math here.*

As mentioned on line 159 in the original submission, we derived Equation 14 directly from (11), not following Eq. 13, though you appear above to also start a derivation from (11). Also in your final sentence, we presume you mean eq.14 not eq.15—the terms 1 and 5 that we had pertain to eq.14 (thus eq.15 remains as previously written, since the product terms to be differentiated are all still within parenthesis; upon using the chain-rule more terms would arise).

We point out that there are several terms lingering in your version that disappear from (14) when picking $S(z) \parallel U(z)$, namely the $V \partial \langle uw \rangle / \partial z$ terms; this can be seen because $|S| \to U$ as well as $V \to 0$. The disappearance of $V$ terms, and also that $|S|^{-1} \to U^{-1}$ cancels out a $U$ (which can be seen more easily before taking the vertical derivative, or alternately after differentiating via recombining), eliminates these terms.

However, there is one term that does not disappear and which we neglected in eq. 14, that we assumed to be negligible: term 4 in the numerator on your last line, $\frac{1}{f|S||G|} \frac{\partial V}{\partial z} \frac{\partial \langle uw \rangle}{\partial z}$. Noting in this coordinate system (i.e., $\mathbf{x} \parallel U(z)$) that (8) means $\partial V / \partial z \to U \partial \varphi / \partial z = |S| \partial \varphi / \partial z$ then this term can be re-written; collecting this on the left-hand side we then have a factor $\left( 1 + \frac{1}{f|G|} \frac{\partial \langle uw \rangle_\parallel}{\partial z} \right) \bigg/ \sqrt{1 - \left( \frac{|S|}{|G|} + \frac{1}{f|G|} \frac{\partial \langle vw \rangle_\perp}{\partial z} \right)^2}$ multiplying the veer (where we use the subscript "$\parallel$" for consistency and to remind of the coordinate system), which dividing out becomes part of the denominator on the right-hand side. Then (14) becomes

$$\frac{\partial \varphi}{\partial z}\bigg|_{\mathbf{e}_x \parallel \mathbf{U}(z)} = \frac{-\frac{|S|}{|G|} \frac{\alpha}{z} - \frac{1}{f|G|} \frac{\partial^2 \langle vw \rangle_\perp}{\partial z^2}}{\sqrt{1 - \left( \frac{|S|}{|G|} + \frac{1}{f|G|} \frac{\partial \langle vw \rangle_\perp}{\partial z} \right)^2} - \frac{1}{f|G|} \frac{\partial \langle uw \rangle_\parallel}{\partial z}} \ ,$$

which is the same as the original submission's (14) but with an additional (small) stress divergence term in the denominator. This has been updated in the text. We point out/remind that this additional term is of similar magnitude as $\frac{1}{f|G|} \frac{\partial \langle vw \rangle_\perp}{\partial z}$; thus (14) is still just a 'competition' between the two terms in the numerator, as written and discussed in the first article draft. This is one reason why eq.38 (now 39) worked as well as it did, as one can see via simple binomial expansion that the denominator can be approximated by $1 - |S|/|G| - \frac{1}{f|G|} \frac{\partial}{\partial z} \left( \langle uw \rangle_\parallel + \langle vw \rangle_\perp \right)$, with $|S|/|G|$ dominant over the stress divergences; this is also seen via the scale analyses in Sect. 2.4 and 3.3.1–3.3.2, due to $c_G$ being between 0.02 and 0.06.

The slight modification/correction to (14) also affects eq. 39 (now eq.40) in section 3.3.1, which has been modified accordingly. We note that due to the limited sensitivity to $h$, the constants in eq.39 (now 40) did not need to be updated when this was modified, and the dashed lines in Fig.15 show negligible difference.

*Also, what is the connection to Equation 10 of the manuscript here. That appears to be far simpler, albeit lacking any shear exponent!*

As already mentioned just below eq. 10 in the text, this relation illustrates that the curvature (second derivative) of stress profiles along with the Coriolis effect are "the basis for mean veer", starting with simple conditions (barotropic, over a homogeneous surface). I.e., it motivates physical understanding, and as noted (eq.10 lacks shear so it isn't "directly useful") it also motivates the more complicated/indirect derivation which immediately follows in section 2.3.

- **Page 10, Line 255:** *Typo in "moreso"? Should be "more so"?*

Corrected.

- *Page 12, Line 294 The velocity profiles based on the shear exponent and the log-law are*

$$|S| = |S|_{\text{ref}} \left( \frac{z}{z_{\text{ref}}} \right)^{\alpha}, \text{ and}$$

$$|S| = \frac{u_*}{\kappa} \ln \left( \frac{z}{z_0} \right)$$

  *respectively. There is quite a bit of discussion on Page 3 Lines $80 - 87$ describing the invalidity of the loglaw above the surface layer and how the shear exponent is better suited for wind energy applications. How do you explain using the log law expression again, especially without the corrections for stability as in M-O law?*

  As discussed in the end of Section 3.3.1 and particularly in the 'Ongoing and future work' part of Sec. 4, various stability-corrected forms were adapted, derived, and used; in the latter section we wrote *"Some relations including stability within $|S|/|G|$ in the shear contribution to veer were developed and tested; however these were not included here, as they did not offer improvement, are seen to be beyond the scope of the current work, and might also require stability effects to be explicitly incorporated within the cross-stress terms. Ongoing work involves addressing the latter: i.e., self-consistent $\alpha$-based description of stability within the veer formulations, within both the shear and cross-stress contribtions in concert with the stability-perturbed geostrophic drag law..."*

- **Page 12, Lines 296-302:** *The estimate for the first vertical derivative of the Reynolds stresses seems reasonable as $\partial \langle uw \rangle / \partial z \approx u_*^2 / h$, although the sign is likely negative. However, the estimate for the second derivative is not very clear. In addition, considering the mean momentum equation from Equation 7,*

$$\frac{\partial \langle vw \rangle}{\partial z} = -f \left( U - U_G \right)$$

$$\frac{\partial^2 \langle vw \rangle}{\partial z^2} = -f \frac{\partial U}{\partial z} = -f \alpha \frac{U}{z}.$$

  *This tells me the sign of this constant $c_{vw}$ must be negative. This might make the numerator of Equation 14 almost zero, suggesting other terms are dominating as expressed in my concerns for Page 7, Line 165. Could you please expand the explanation for your estimate of the second derivative of the stresses?*

  The sign of $\partial \langle uw \rangle / \partial z$ is not negative, as seen in e.g. the textbook by Wyngaard (2010), which we reference; we remind that $\langle uw \rangle$ begins as $u_*^2$ at the surface, increasing towards zero (magnitude decreasing with $z$) as one moves away from the surface; thus $\partial \langle uw \rangle / \partial z$ is positive ($\approx u_*^2 / h$, as written). Indeed the sign of $c_{vw}$ is negative, as we had stated on lines 612–613 (*"Given the negative curvature of lateral stress, $\partial^2 \langle vw \rangle / \partial z^2 < 0$ (Wyngaard, 2010), $c_{vw}$ is negative and of order 1..."*) As written on line 617, *"The value of $c_{vw}$ giving the estimates shown in Fig. 15 was $-0.7$ for all sites"*. The estimate of the stress curvature (second derivative) was explained and given on lines 299–304 as well as eq.34.

- **Page 12, Line 306:** *Based on the expressions in Equation 31, I got this to be*

$$\sin \varphi_G = \frac{-c_G}{\kappa} \left\{ B \cos \varphi_0 - \left[ \ln \frac{u_*}{f z_0} - A \right] \sin \varphi_0 \right\}$$

  *The multiplication of $B$ by $\cos \varphi_0$ is missing. Could you please check again.*

  We agree: checking again we see the factor of $\cos \varphi_0$ on $B$ was omitted when transferring from handwritten to LaTeX; this has been corrected.

- **Page 12, Line 315:** *A summary of the expressions to be evaluated after substitutions listed in Section 2.4 would be extremely helpful before evaluating them in Section 3.1.*

  To limit the length of the paper we have not added more expressions, preferring to refer to them directly when using them to make the practical/applied equations 38–39 (now eq.39–40).

- **_Page 14, Line 355_**: _A more thorough overview of how the RANS simulations covers the space of interest would be helpful here._

  Lines 349–351 (now l.392–394 in updated manuscript) also described how the RANS simulations cover the space of interest. Further, Fig. 6c visually displays this also in terms of shear and veer.

- **_Page 14, Line 358_**: _How does the expression in Equation 14 compare to Equation 9 in the right plot in Figure 2? Based on my concerns expressed for Page 7, Line 165, I expect the correlation to be not as good as for Equation 9._

  As mentioned in our response above, inclusion of the lateral shear piece introduces a (small) modification into the denominator of (14); since it is in the denominator it does not affect the (numerator) components shown in Fig. 2 nor the balance shown in Fig. 5. Thus the correlation is just as good.

- **_Page 15, Line 374_**: _Behavior that is "not inconsistent"? Do you mean consistent?_

  As addressed in the authors' reply to reviewer 1 ("AC1" on 21 Mar.2023), this has been changed to 'consistent.'

  _There is no expression derived in Section 2 that gives $Ro_h^{1.4}$. This is also likely affected by concerns for Page 7, Line 165._

  We have now specifically noted _"the veer is empirically found to be proportional to $Ro_h^{1.4}$..."_ This is not connected to eq.14 nor its derivation (your concerns around line 165), but rather comes directly from the RANS output which are independent of such derivation (they just solve the mean Navier-Stokes equations).

- **_Page 26, Line 583 and Page 28, Line 609_**: _This simplification in Equations 38 and 39 are likely in error. Could you please check my concerns for Page 7, Line 165 before confirming these results._

  As mentioned above addressing your previous concerns: the simplification in eq.38 (now 39) was not affected; the small change to eq.39 (now 40), which we updated in the manuscript, did not cause a noticeable difference.

- **_Page 29, Figure 16_**: _The standard deviation of the veer is almost the same or much larger than the mean from the experiments. It would be hard to consider this as proof of the expressions derived in Equations 38 and 39._

  These are two different things, and remind that the simplified expressions are providing an $\alpha$-based estimate for the mean. The standard deviation conditioned on shear exponent $\sigma_{\Delta\varphi/\Delta z|\alpha}$ was discussed in lines 635–645 (now 683–693) and shown in Fig. 16. We considered developing an empirical model for $\sigma_{\Delta\varphi/\Delta z|\alpha}$, but deemed this to be beyond the scope of the current paper and is ongoing work. We also note that we gave a form to estimate the standard deviation of veer conditioned on wind speed, which was eq.37 (now 38).

Again we express appreciation to the Ganesh Vijayakumar for the review and his checking of the derivations, which helped to both improve the article as well as support the scientific method.

with kind regards,
    Mark Kelly and M. Paul van der Laan
    Department of Wind and Energy Systems
    Risø Lab/Campus, Danish Technical University